# The RAD51 recombinase protects mitotic chromatin in human cells

Isabel E. Wassing[1], Emily Graham[1], Xanita Saayman[1,3], Lucia Rampazzo[1,3], Christine Ralf[1], Andrew Bassett [2] &
Fumiko Esashi [1✉]

The RAD51 recombinase plays critical roles in safeguarding genome integrity, which is fundamentally important for all living cells. While interphase functions of RAD51 in maintaining genome stability are well-characterised, its role in mitosis remains contentious. In this study, we show that RAD51 protects under-replicated DNA in mitotic human cells and, in this way, promotes mitotic DNA synthesis (MiDAS) and successful chromosome segregation. In cells experiencing mild replication stress, MiDAS was detected irrespective of mitotically generated DNA damage. MiDAS broadly required de novo RAD51 recruitment to single-stranded DNA, which was supported by the phosphorylation of RAD51 by the key mitotic regulator Polo-like kinase 1. Importantly, acute inhibition of MiDAS delayed anaphase onset and induced centromere fragility, suggesting a mechanism that prevents the satisfaction of the spindle assembly checkpoint while chromosomal replication remains incomplete. This study hence identifies an unexpected function of RAD51 in promoting genomic stability in mitosis.

[1] Sir William Dunn School of Pathology, University of Oxford, Oxford, UK. [2] Wellcome Sanger Institute, Cambridge, UK. [3]These authors contributed equally:
Xanita Saayman, Lucia Rampazzo. ✉email: fumiko.esashi@path.ox.ac.uk

The RAD51 recombinase is essential for cell survival and for the protection of genome stability[1]. RAD51 is best known as the central catalyst of homologous recombination (HR), which provides error-free repair of double-stranded DNA breaks (DSBs) during S and G2 phases of the cell cycle. In this process, DSBs are first recognised by the MRE11-RAD50-NBS1 (MRN) complex, which initiates resection of the broken DNA ends[2]. The resected 3' single-stranded DNA (ssDNA) overhang is rapidly bound by the replication protein A (RPA) and subsequently replaced by RAD51. The resulting RAD51 nucleoprotein filament mediates strand invasion to initiate HR repair[3]. RAD51 similarly promotes replication restart through break-induced replication (BIR) at broken replication forks[4–6]. Additionally, RAD51 plays non-repair functions at stalled replication forks, in which RAD51 stimulates fork reversal and stabilises stalled forks by protecting against the nucleolytic degradation of nascent DNA[7–12].

In interphase, RAD51 loading at resected DSBs is largely mediated by the breast cancer susceptibility 2 protein (BRCA2)[13–15]. BRCA2 is recruited to DSB sites through its interaction with the partner and localiser of BRCA2 (PALB2) and the breast cancer susceptibility 1 protein (BRCA1)[16]. Significantly, our previous studies uncovered a separate mechanism that enables RAD51 recruitment to damaged DNA independently of BRCA2 chromatin enrichment[17,18]. This pathway is triggered by a key mitotic regulator, Polo-like kinase 1 (PLK1), which phosphorylates RAD51 at serine 14 (S14) shortly after DNA damage and in later phases of the cell cycle. This event subsequently provokes RAD51 phosphorylation by the acidophilic casein kinase 2 (CK2) at threonine 13 (T13), which mediates a direct interaction between RAD51 and the NBS1 component of the MRN complex. In this way, PLK1 and CK2-mediated RAD51 phosphorylation allows RAD51 to be recruited to broken DNA ends and also to stalled replication forks with extended ssDNA[17]. Given that RAD51 S14/T13 phosphorylation peaks during late G2 phase and mitosis in unperturbed cells[17,18], these observations raise the intriguing possibility that, similar to the well-described role of RAD51 in meiosis[19], RAD51 acts on mitotic chromatin at sites of DNA damage or extended ssDNA regions.

Notably, emerging evidence demonstrates that, under conditions of mild replicative stress, cancer cells enter mitosis before the completion of DNA replication[20–22]. The persistence of under-replicated DNA in mitosis, often described at common fragile sites (CFSs), leads to impaired chromosome disjunction, observed as anaphase bridges, ultra-fine bridges (UFBs) and micronuclei[20,22–25]. Several DNA repair proteins, such as FANCD2, MUS81, TOPBP1, and TLS polymerases Pol η (eta) and Pol ζ (zeta), have been found to act at CFSs to alleviate these phenotypes[20–22,25–28]. It has been proposed that under-replicated CFSs complete replication in early mitosis via a BIR-like mechanism termed mitotic DNA synthesis (MiDAS), which is triggered by nucleolytic cleavage of stalled replication forks, hence DSB induction, upon mitotic entry[24,29,30]. While several reports in higher eukaryotes suggest that RAD51 catalyses BIR, MiDAS at CFSs appears to be a RAD52-dependent process[4,6,31,32]. Nonetheless, a complete picture of the mechanisms orchestrating MiDAS is currently lacking.

Given that cancer cells generally cope with increased levels of oncogene-induced replication stress, MiDAS is thought to be particularly important for their survival, making the inhibition of MiDAS an attractive avenue for cancer therapy[31]. Indeed, the detrimental consequences of an under-replicated genome are indicated by the increase in chromosome mis-segregation and non-disjunction observed upon MiDAS inhibition[24]. Chromosome segregation during mitosis is centrally controlled by the spindle assembly checkpoint (SAC), which prevents anaphase onset until the mitotic spindles correctly attach to the kinetochore machinery assembled on the centromere of each replicated sister chromatid[33]. Importantly, the centromere is a late-replicating region of the genome, harbouring unusually long stretches of repetitive DNA, which is susceptible to a high level of recombination[34,35]. It remains poorly understood, however, how DNA synthesis in mitosis impacts on the integrity of centromere DNA repeats, SAC maintenance and subsequent events of chromosome segregation.

In this study, we investigated the potential involvement of RAD51 in MiDAS and its impact on mitotic progression. We show that RAD51 promotes MiDAS irrespective of the induction of DNA damage upon mitotic entry, and that this involves RAD51-mediated protection of ssDNA. Analysis of RAD51 missense mutants at S14 further indicates that RAD51 phosphorylation by PLK1 mediates this mitotic RAD51 function. Additionally, we describe a relationship between the timely completion of DNA replication and SAC signalling. Cells that exhibit inherently high levels of replication stress displayed increased cellular death upon SAC inhibition. Furthermore, MiDAS inhibition delays the onset of anaphase and induces increased fragility at the centromeres, providing a mechanistic insight into the maintenance of SAC signalling in under-replicated cells. Taken together, we propose that RAD51 promotes MiDAS to ensure completion of replication at under-replicated DNA including centromeres and, in this way, furthers mitotic progression and accurate chromosome inheritance.

## Results

**MiDAS in U2OS cells is largely associated with unbroken DNA.** We first evaluated the nature of DNA synthesis detectable upon mitotic release, formally defined as MiDAS[24], in further detail. To this end, U2OS cells were synchronised at G1/S by double thymidine block and released into S phase (dT-B/R) in medium containing a low dose (0.4 μM) of aphidicolin (APH), an inhibitor of B-family DNA polymerases α, δ and ε. CDK1 inhibitor RO3306 was then added as the cells entered G2 phase to arrest them at the G2/M boundary, followed by mitotic release (m-R) into fresh medium containing the thymidine analogue EdU (Supplementary Fig. 1a). Cells undergoing mitotic rounding were then collected by mitotic shake-off (m-SO) and EdU incorporation was analysed by fluorescent click-labelling (EdU-Click-F). In proof-of-principle experiments (Supplementary Fig. 1b and c), we addressed the possibility that EdU incorporation in late G2 (prior to m-R) contributes to the resultant EdU foci detected in mitotically collected cells by administrating the EdU pulse 30 min prior to m-R with or without a subsequent chasing with unlabelled thymidine. While a substantial amount of EdU signal was observed in G2 pulse-labelled cells (G2), this signal was drastically reduced when the G2-pulse was followed by a thymidine chase upon m-R (G2 + Thy). Given that the thymidine chase outcompetes the incorporation of trace EdU amounts remaining inside of cells after m-R, this observation demonstrates that the EdU signal observed in G2 pulse-labelled cells mostly represents EdU incorporation after m-R, and that negligible EdU incorporation occurs during the G2-pulse itself. Conversely, strong EdU signals were detectable when cells were immediately exposed to EdU at m-R (M 0'). However, when EdU addition was delayed for 5 min or longer after m-R (M 5', 10' or 15'), EdU incorporation significantly decreased. Together, these observations validate mitotic EdU foci as a read-out for MiDAS, which predominantly occurs in the very early stages of mitosis.

We next examined whether the duration of RO3306 incubation affected MiDAS (Supplementary Fig. 1d). Similar to previous reports[36], a significant reduction of mitotic EdU incorporation was detected when cells experienced a longer RO3306-arrest (Supplementary Fig. 1e), supporting the view that APH-induced under-

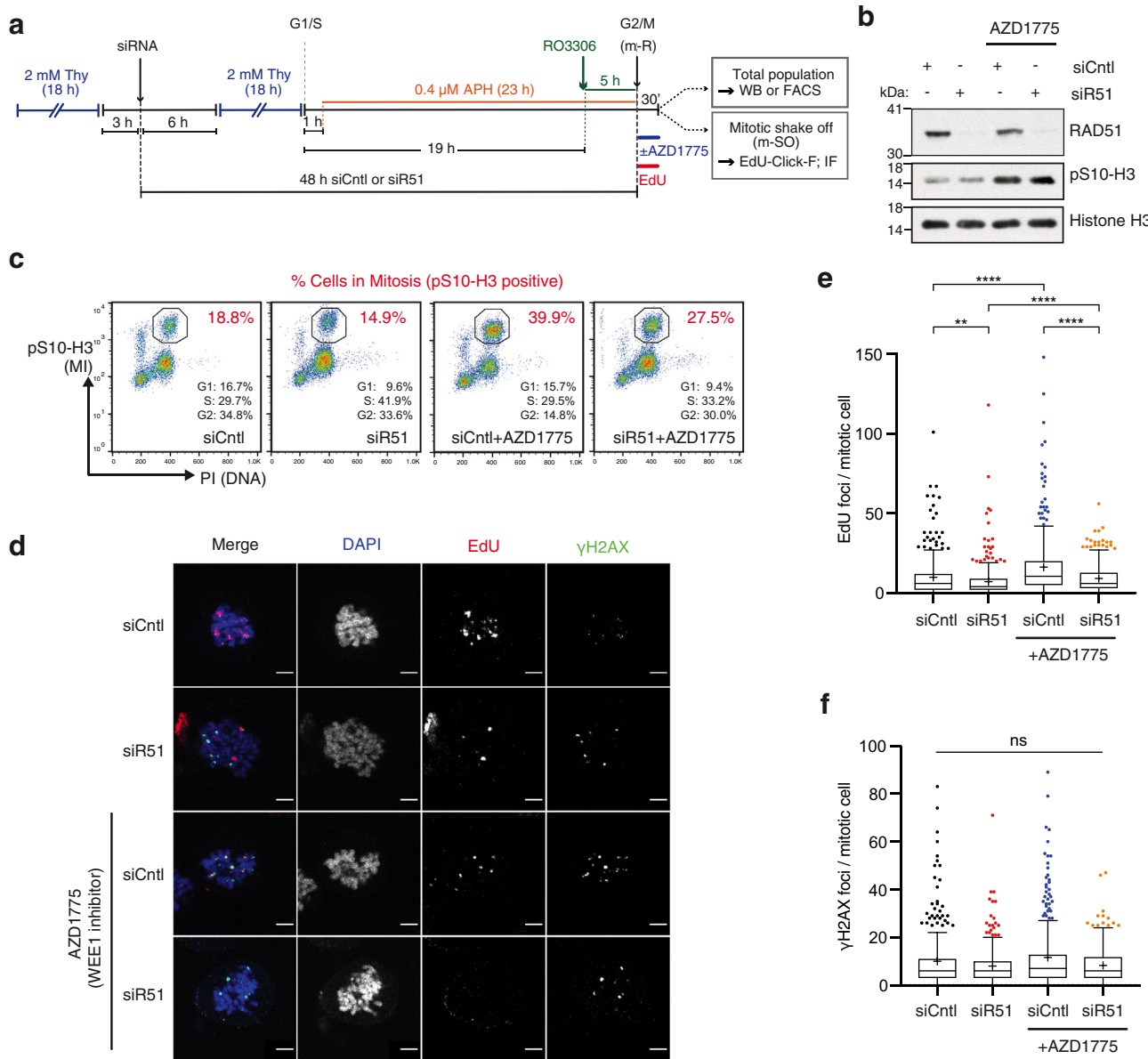

**Fig. 1 RAD51 depletion significantly reduces MiDAS. a** Schematic diagram of U2OS cell synchronisation by double thymidine block and release (dT-B/R) with siRNA treatment against RAD51 (siR51) or control (siCntl), followed by exposure to low-dose aphidicolin (0.4 μM APH) and G2/M arrest by CDK1 inhibitor (9 μM, RO3306). Cells were released into mitosis (m-R) in medium containing 10 μM EdU with or without 100 nM WEE1 inhibitor (100 nM, AZD1775). **b**, **c** Western blot analysis against RAD51, mitotic marker pS10-H3 and Histone H3 (**b**) or FACS analysis for mitotic index (MI) (pS10-H3) and DNA content (PI) (**c**) of the total cell population 30 min after m-R, according to the workflow depicted in **a**. Based on a single experiment. **d** Representative images of cells collected by mitotic shake-off (m-SO) at 30 min after m-R, stained for EdU incorporation and γ-H2AX and RPA. Scale bar indicates 5 μm. **e**, **f** Quantification of the number of EdU foci (**e**) and γ-H2AX foci (**f**) per mitotic cell. Data were obtained from three independent experiments (100 cells analysed per repeat); n = 300 per condition. Data distribution is represented by Tukey box-and-whisker plots. Bounds of box are 25–75th percentile, centre shows the median and '+' marks the mean. Whiskers indicate ±1.5xIQR, data outside this range are drawn as individual dots. p-values were calculated by a Kruskal–Wallis test followed by Dunn's multiple comparison test. Asterisks indicate **p-value ≤ 0.01; ****p-value ≤ 0.0001. Source data are provided as a Source Data file.

replication can, at least in part, be resolved by extending the duration of G2. Mitotic γ-H2AX, a widely used marker of the DNA damage response (DDR), was similarly reduced upon prolonged RO3306-arrest (Supplementary Fig. 1f). Nonetheless, a substantial amount of mitotic EdU incorporation remained even after 12 h of RO3306 incubation, suggesting that a subset of under-replicated regions require release from RO3306, or mitotic entry, to be fully resolved. Intriguingly, we noticed that, for all the different synchronisation protocols tested for U2OS cells, only a subset of EdU foci (<30%) colocalised with γ-H2AX (Supplementary Fig. 1 g).

In HEK293 cells, in contrast, the majority of mitotic EdU foci (64%) colocalised with γ-H2AX (Supplementary Fig. 1h). Importantly, while γ-H2AX is commonly considered to be associated with DSBs, γ-H2AX foci can also be formed upon other types of genotoxic stress in the absence of a DSB and can remain for several hours after the completion of DNA repair[37,38]. Mitotic EdU foci that colocalise with γ-H2AX therefore mark MiDAS events either at broken replication forks or at otherwise stressed replication forks, referred to as DDR-associated loci, whereas those that do not colocalise with γ-H2AX indicate previously unrecognised MiDAS

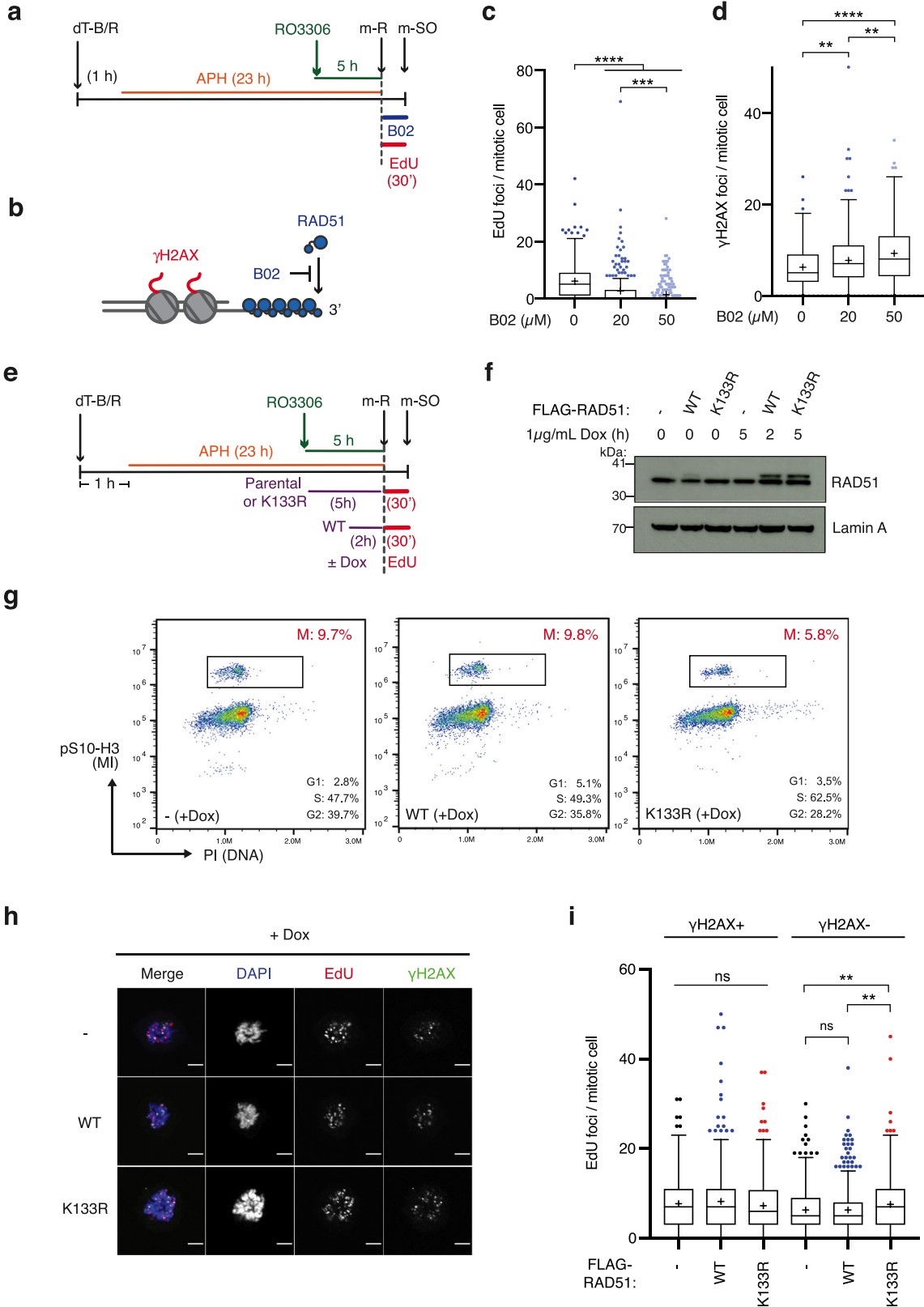

events at chromatin loci that are undamaged or escaped from canonical DDR, referred to as non-DDR-associated loci. Overall, these observations suggest that the nature of MiDAS varies significantly between different cell lines; MiDAS is largely associated with non-DDR loci in U2OS, but predominantly occurs at DDR-associated loci in HEK293 cells.

**RAD51 promotes MiDAS**. We next set out to investigate the potential involvement of RAD51 in MiDAS. To this end, U2OS cells were synchronised in combination with siRNA treatment such that mitotic cells were collected 48 h after transfection (Fig. 1a). Owing to its essential role during DNA replication, prolonged RAD51 depletion was expected to trigger G2 arrest[39,40], thereby preventing mitotic

**Fig. 2 RAD51 association with mitotic chromatin promotes MiDAS. a** Schematic diagram of experimental procedure in which cells were synchronised by double thymidine block and release (dT-B/R) and G2/M arrest by RO3306 treatment, and exposed to small molecule inhibitors of RAD51 (B02) or vehicle (-) upon mitotic release (m-R). Subsequently, mitotic cells were collected by mitotic shake-off (m-SO). **b** Depiction of RAD51 association with resected ssDNA and associated $\gamma$-H2AX, spanning an estimated 1–2 Mb. B02 blocks de novo RAD51 loading but does not disrupt a pre-formed filament. **c, d** Mitotic cells collected by m-SO, following 10 $\mu$M EdU exposure with or without B02 at the indicated concentrations, were stained and quantified for EdU foci (**c**) or $\gamma$-H2AX foci (**d**) per mitotic cell. Data were obtained from three independent experiments (100 cells analysed per repeat); $n = 300$ per condition. Data distribution is represented by Tukey box-and-whisker plots. Bounds of box are 25–75th percentile, centre shows the median and '+' marks the mean. Whiskers indicate ±1.5xIQR, data outside this range are drawn as individual dots. $p$-values were calculated by a Kruskal–Wallis test followed by Dunn's multiple comparison test. Asterisks indicate **$p$-value ≤ 0.01; ***$p$-value ≤ 0.001; ****$p$-value ≤ 0.0001. **e** Schematic diagram of experimental procedure to synchronise U2OS Flp-In T-REx cell lines expressing FLAG-tagged RAD51 variants from the tet-O promoter. Cells were synchronised in the presence of low-dose APH, with or without the addition of doxycycline as indicated for each cell line. **f** Western blot analysis of Lamin A (loading control) and RAD51 protein levels for the total cell population synchronised according to the workflow depicted in **e**. Based on a single experiment. **g** FACS analysis of mitotic index (MI) (pS10-H3) and DNA content (PI) following cell synchronisation according to the experimental procedure depicted in Fig. 2e. At 30 min after m-R, the total cell population was collected by trypsinization and analysed by FACS. The G1, S and G2 populations were estimated by the Watson Pragmatic algorithm (FlowJo), based on the PI staining of interphase cells. **h** Representative images of mitotic cells, stained for EdU and $\gamma$-H2AX, quantified in (**i**) and Supplementary Fig. 2e and f. Scale bar indicates 5 $\mu$m. **i** Quantification of the number of EdU foci that colocalise with $\gamma$-H2AX ($\gamma$-H2AX $^+$) and the number of EdU foci that do not colocalise with $\gamma$-H2AX ($\gamma$-H2AX$^-$) per mitotic cell in doxycycline treated cells. Data were obtained from four independent experiments; $n = 400$ per condition. Data distribution is represented by Tukey box-and-whisker plots. Bounds of box are 25–75th percentile, centre shows the median and '+' marks the mean. Whiskers indicate ±1.5xIQR, data outside this range are drawn as individual dots. $p$-values were calculated by a Kruskal–Wallis test followed by Dunn's multiple comparison test. Asterisks indicate **$p$-value ≤ 0.01. Source data are provided as a Source Data file.

entry of damaged or heavily under-replicated cells, hence potentially precluding the unbiased analysis of MiDAS. To override this G2 arrest, cells were co-treated with the WEE1 inhibitor AZD1775 (previously known as MK-1775) upon m-R. Western blot analysis confirmed the successful depletion of RAD51 from the total cell population (Fig. 1b). As expected, cells treated with siRNA targeting RAD51 (siR51) showed a reduced mitotic population compared to cells treated with a negative control siRNA (siCntl), and mitotic entry was increased upon WEE1 inhibition (Fig. 1c). We found that mitotic EdU incorporation was significantly decreased in RAD51-depleted cells compared to control cells, while the number of mitotic $\gamma$-H2AX foci were unchanged (Fig. 1d–f). This trend remained in cells treated with WEE1 inhibitor. Together, the observed reduction in mitotic EdU incorporation upon siRNA-mediated RAD51 depletion likely indicates the impairment of MiDAS.

In order to confirm the importance of RAD51 and its association with chromatin in MiDAS, we used a well-characterised small molecule inhibitor of RAD51 (B02) to rapidly block RAD51 chromatin recruitment in mitosis without impacting RAD51 function during the preceding S phase[41] (Fig. 2a). Previous biochemical analyses have shown that B02 inhibits both RAD51 binding to ssDNA and the RAD51-driven strand invasion process, but does not disrupt an already established RAD51 nucleoprotein filament[42]. Corroborating this reported impact of B02, damage-induced RAD51 foci formation was significantly hindered when B02 was added prior to, but not after, irradiation (Supplementary Fig. 2a–c). Therefore, as depicted in Fig. 2b, B02 specifically inhibits de novo RAD51 recruitment to DNA without impacting existing RAD51 filaments. We found that addition of B02 upon m-R drastically reduced mitotic EdU incorporation (Fig. 2c), indicating that de novo RAD51 recruitment to mitotic chromatin indeed promotes MiDAS. A corresponding increase in $\gamma$-H2AX foci was observed upon mitotic RAD51 inhibition (Fig. 2d), indicating the importance of de novo RAD51 recruitment in preventing the accumulation of DNA damage during mitosis.

To further evaluate the significance of stable RAD51 association with DNA in mitosis, we exploited the ATPase-defective RAD51 K133R variant that associates with DNA with high affinity. Exogenous expression of the RAD51 K133R mutant stabilises RAD51 filaments at ssDNA, thereby enhancing RAD51-mediated ssDNA protection[10]. The K133R variant also retains

strand exchange activity, although its expression in human cells impairs the efficient completion of HR repair due to a defect in filament disassembly[7,8,43]. We generated U2OS Flp-In T-REx cell lines in which the expression of FLAG-tagged wild-type (WT) RAD51 or the K133R variant can be induced upon doxycycline exposure. To achieve similar levels of exogenous RAD51 expression, doxycycline was added 5 h prior to m-R for the RAD51 K133R-expressing cell line and 2 h prior to m-R for the RAD51 WT-expressing cell line (Fig. 2e, f and Supplementary Fig. 2d). Under these conditions, RAD51 K133R-expressing cells exhibited some delays in replication (Fig. 2g), potentially reflecting the negative impact of stable RAD51 association with DNA. Nonetheless, when mitotic cells are analysed for MiDAS, we found that RAD51 K133R expression conferred a modest but significant increase in mitotic EdU foci at $\gamma$-H2AX-negative loci, but not at $\gamma$-H2AX-positive loci, compared to the parental and RAD51 WT-expressing cell lines (Fig. 2h, i and Supplementary Fig. 2e, f). RAD51 K133R expression also decreased overall $\gamma$-H2AX foci compared to RAD51 WT-expressing cells (Supplementary Fig. 2f), suggesting that enhanced RAD51 chromatin association protects against mitotic DNA damage and, in this way, directs under-replicated foci towards non-DDR-associated MiDAS. Collectively, these observations suggest that mitotic RAD51 recruitment promotes MiDAS and that this involves RAD51-mediated ssDNA protection at non-DDR-associated loci.

**MiDAS at non-DDR-associated loci is specifically promoted by RAD51.** MiDAS is widely considered to be a RAD52- and POLD3-dependent process[24,31] and predominantly occurs at CFSs and also at telomeres via a BIR-like mechanism, hence constituting DDR-associated loci[29,30,32]. To better understand the relative contributions of these factors in the wider context of globally detected MiDAS, we compared the effects of RAD51-, RAD52- and B-family polymerase inhibition using B02, AICAR and APH, respectively (Fig. 3a). As shown in Fig. 3b–d, mitotic RAD51 inhibition drastically reduced MiDAS with a significant increase in mitotic $\gamma$-H2AX foci compared to vehicle (DMSO) treated control cells. RAD52 inhibition also elicited a reduction in mitotic EdU foci, albeit at a relatively modest level. Under this condition, however, we found no increase in $\gamma$-H2AX foci compared to control cells. The addition of high-dose (2 $\mu$M) APH in mitosis (APH-M), on the other hand, fully abolished MiDAS,

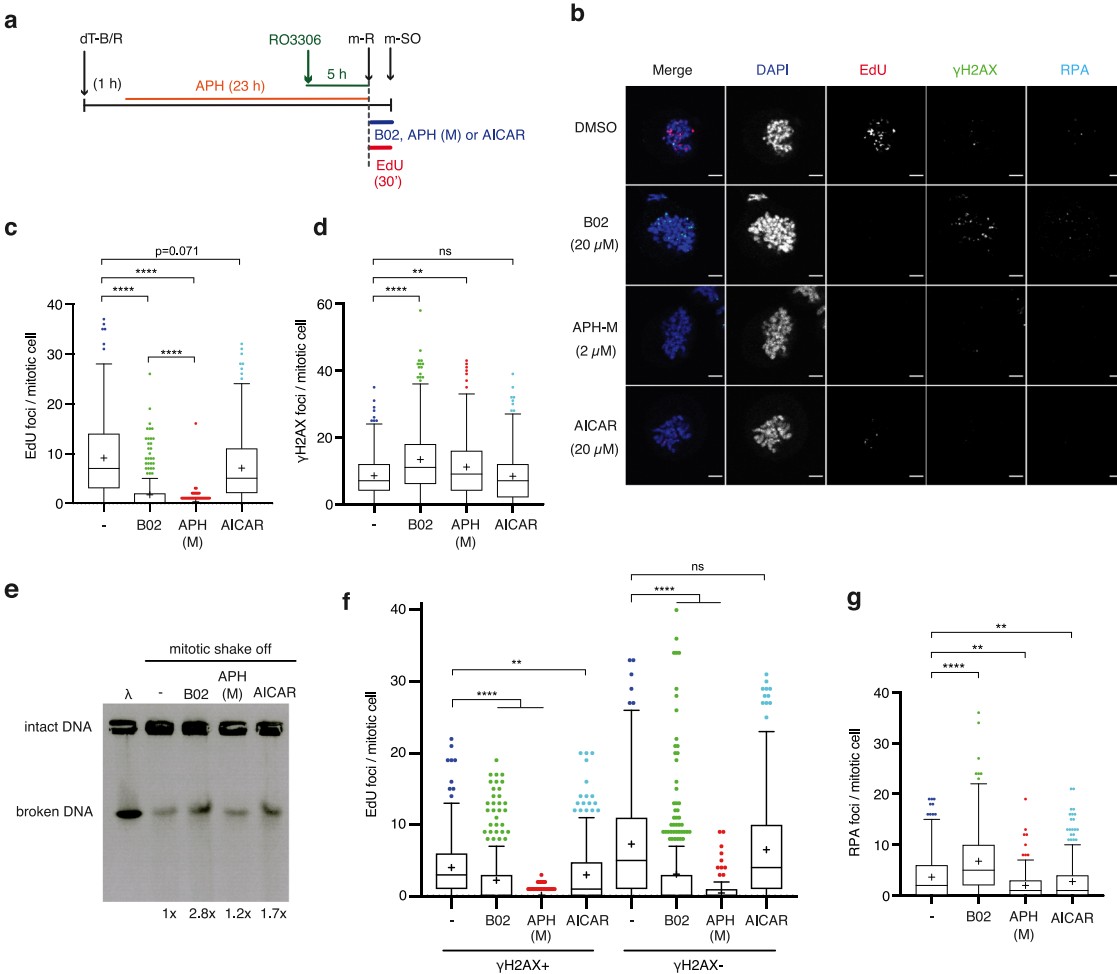

**Fig. 3 RAD51 depletion in mitosis induces DNA breakages. a** Schematic diagram of experimental procedure in which U2OS cells were synchronised in the presence of low-dose APH and exposed to vehicle (-) or small molecule inhibitors of RAD51 (B02), B-family polymerases (2 μM APH (M)) or RAD52 (20 μM AICAR) upon mitotic release (m-R). **b** Representative images of mitotic cells collected by mitotic shake-off (m-SO), following 10 μM EdU exposure with or without indicated small molecule inhibitors, stained for EdU incorporation and γ-H2AX and RPA. Scale bar indicates 5 μm. **c, d**. Quantification of the number of EdU foci (**c**) and γ-H2AX foci (**d**) per mitotic cell. Data were obtained from three independent experiments (100 cells analysed per repeat); n = 300 per condition. Data distribution is represented by Tukey box-and-whisker plots. Bounds of box are 25–75th percentile, centre shows the median and '+' marks the mean. Whiskers indicate ±1.5xIQR, data outside this range are drawn as individual dots. *p*-values were calculated by a Kruskal–Wallis test followed by Dunn's multiple comparison test. Asterisks indicate **\**p*-value ≤ 0.01; \*\*\*\**p*-value ≤ 0.0001. **e** Pulsed-field Gel Electrophoresis (PFGE) of mitotic cells, collected by m-SO, after synchronisation and the indicated drug treatments upon m-R. DNA fragments of 1–100 kb size were separated by electrophoresis. Lambda (λ) DNA ladder was used as DNA size standard. Total band intensity was normalised to untreated (-) control (Quantification). Based on a single experiment. **f** Quantification of the number of EdU foci that colocalise with γ-H2AX (γ-H2AX + EdU foci) and the number of EdU foci that do not colocalise with γ-H2AX (γ-H2AX⁻ EdU foci) per mitotic cell. Data were obtained from three independent experiments (100 cells analysed per repeat); n = 300 per condition. Data distribution is represented by Tukey box-and-whisker plots. Bounds of box are 25–75th percentile, centre shows the median and '+' marks the mean. Whiskers indicate ±1.5xIQR, data outside this range are drawn as individual dots. *p*-values were calculated by a Kruskal–Wallis test followed by Dunn's multiple comparison test. Asterisks indicate **\**p*-value ≤ 0.01; \*\*\*\**p*-value ≤ 0.0001. **g** Quantification of the number of RPA foci per mitotic cell. Data were obtained from three independent experiments (100 cells analysed per repeat); n = 300 per condition. Data distribution is represented by Tukey box-and-whisker plots. Bounds of box are 25–75th percentile, centre shows the median and '+' marks the mean. Whiskers indicate ±1.5xIQR, data outside this range are drawn as individual dots. *p*-values were calculated by a Kruskal–Wallis test followed by Dunn's multiple comparison test. Asterisks indicate **\**p*-value ≤ 0.01; \*\*\*\**p*-value ≤ 0.0001. Source data are provided as a Source Data file.

which was accompanied by a modest increase in mitotic γ-H2AX foci. The greater contribution of RAD51 to MiDAS compared to RAD52 was further corroborated by siRNA analysis (Supplementary Fig. 3b–f).

As γ-H2AX does not always represent physical DNA breaks, we also conducted pulsed-field gel electrophoresis (PFGE) to assess the presence of DSBs in mitotically collected cells (Fig. 3e). PFGE analysis of B02-treated mitotic cells revealed a 2.8-fold increase of broken DNA, detected as a fast migrating form,

compared to the vehicle-treated control. This observation agrees with the increase of γ-H2AX foci in B02-treated cells (Fig. 3d), indicating the enhanced induction of mitotic DSBs and thus corroborating that RAD51 protects ssDNA in mitosis. While AICAR treatment did not alter the number of mitotic γ-H2AX foci (Fig. 3d), we observed a substantial increase in broken DNA (1.7 fold) by PFGE (Fig. 3e). The discrepancy between PFGE (physical DNA breaks) and γ-H2AX foci analysis (DDR marker) in AICAR-treated cells can be explained by the fact that γ-H2AX

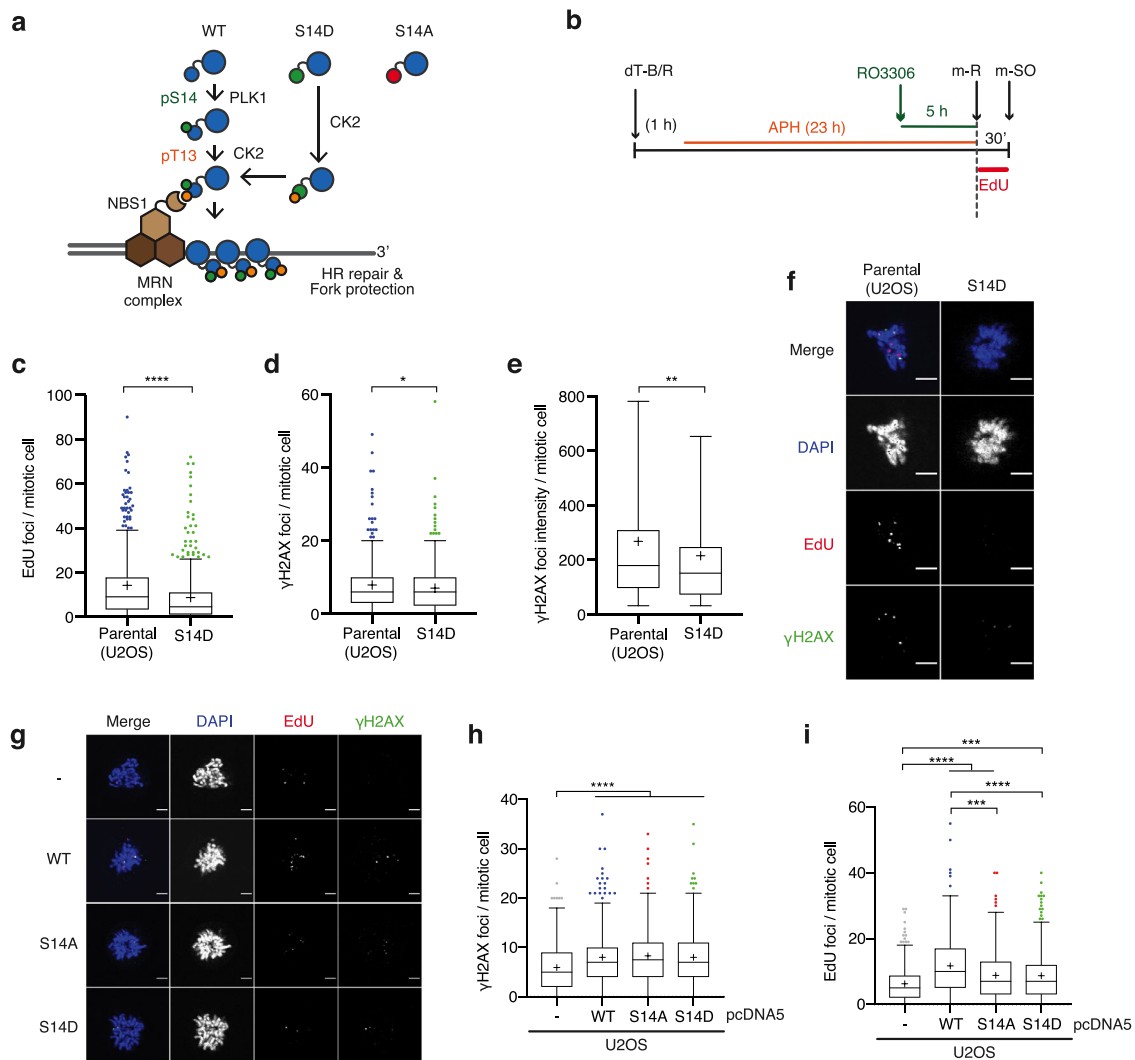

**Fig. 4 PLK1-dependent RAD51 phosphorylation promotes non-DDR-associated MiDAS. a** Schematic illustration of the mechanism recruiting RAD51 to sites of DNA damage via PLK1-dependent RAD51 S14 phosphorylation, triggering CK2-dependent RAD51 phosphorylation at T13, which mediates NBS1 interaction. **b** Schematic diagram of experimental procedure used to obtain data in Fig. 4c–i and Supplementary Fig. 5c–e. **c–e** EdU incorporation and *γ*-H2AX IF signal were analysed in the parental U2OS cell line and the U2OS *RAD51^{S14D/S14D}* mutant cell line (S14D) following cell synchronisation as depicted in **b**. Quantification of the number of EdU foci (**c**) and the number of *γ*-H2AX foci per mitotic cell (**d**). Quantification of the average *γ*-H2AX focus intensity per mitotic cell (**e**), for the same data as (**d**). Data were obtained from four independent experiments (100 cells analysed per repeat); *n* = 400 per condition. Data distribution in **c** and **d** is represented by Tukey box-and-whisker plots. Bounds of box are 25–75th percentile, centre shows the median and '+' marks the mean. Whiskers indicate ±1.5xIQR, data outside this range are drawn as individual dots. Data distribution in **e** is represented by a box-and-whisker plot where the bounds of the box are 25–75th percentile, centre shows the median and '+' marks the mean. Whiskers indicate 5-95 percentile, data outside this range is not shown. Significant differences between samples were determined by a two-sided Mann–Whitney test. Asterisks indicate *\*p*-value ≤ 0.05; \*\**p*-value ≤ 0.01; \*\*\*\**p*-value ≤ 0.0001. **f** Representative images of the data analysed in **i–e** are shown. Scale bar indicates 5 μm. **g–i** EdU incorporation and *γ*-H2AX IF signal were analysed in U2OS cells stably carrying pcDNA5 vector (-); or cDNA encoding wild-type RAD51 (WT), RAD51 S14A or RAD51 S14D mutants. Representative images of the data analysed (**g**). Scale bar indicates 5 μm. Quantification of the number of *γ*-H2AX foci (**h**) and the number of EdU foci per mitotic cell (**i**). Data were obtained from three independent experiments (100 cells analysed per repeat); *n* = 300 per condition. Data distribution are represented by Tukey box-and-whisker plots. Bounds of box are 25–75th percentile, centre shows the median and '+' marks the mean. Whiskers indicate ±1.5xIQR, data outside this range are drawn as individual dots. *p*-values were calculated by a Kruskal–Wallis test followed by Dunn's multiple comparison test. Asterisks indicate \*\*\**p*-value ≤ 0.001; \*\*\*\**p*-value ≤ 0.0001. Source data are provided as a Source Data file.

removal occurs with slower kinetics compared to the DSB repair[38], such that *γ*-H2AX foci persist regardless of whether the mitotically induced DSB has been repaired by RAD52 shortly after mitotic entry (within 30 min). Defective mitotic break repair in RAD52 inhibited cells therefore results in an increase of unrepaired mitotic DSBs, which can be detected by PFGE, without affecting the number of mitotic *γ*-H2AX foci. The phenotypes observed in AICAR-treated cells hence corroborate that RAD52

catalyses MiDAS at broken DNA. Conversely, PFGE analysis of APH-M treated cells showed only a marginal increase of the fast migrating form of DNA (1.2-fold). Under this condition, RAD51 and RAD52 are both functional, such that APH-induced DNA polymerase stalling likely elicits the accumulation of unresolved DNA repair-intermediates. Such DNA structures may fail to migrate into the agarose matrix, as has been shown for branched DNA replication intermediates[38,44], explaining the PFGE result. Models

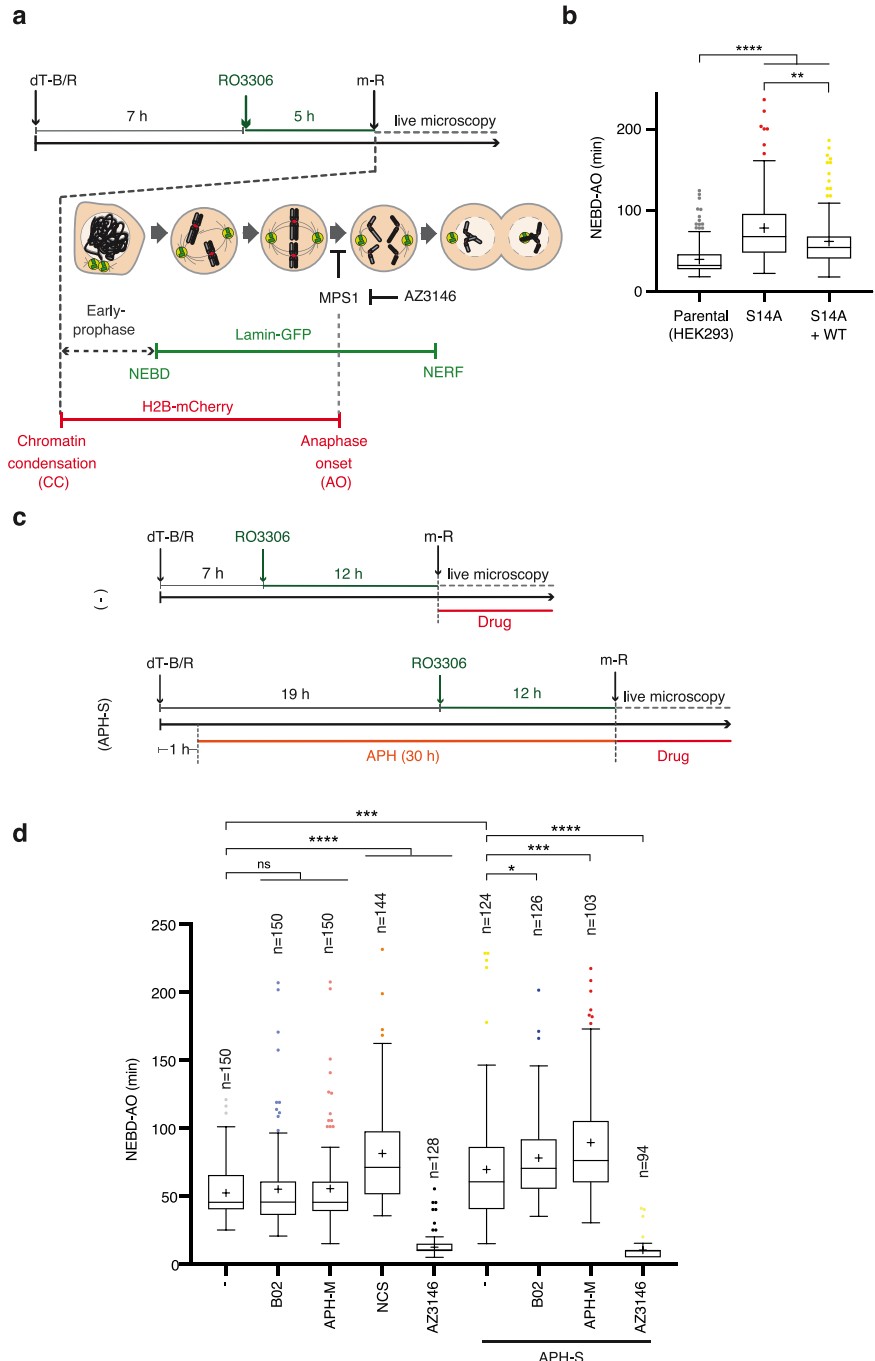

describing the various phenotypes observed upon mitotic B02, AICAR or APH-M treatment are depicted in Supplementary Fig. 3a.

To further evaluate the differential contribution of RAD52 and RAD51 in DDR-associated MiDAS and non-DDR-associated MiDAS, we assessed EdU foci colocalizing with and without $\gamma$-H2AX. Consistent with previous reports, we found that the impact of RAD52 inhibition was limited to EdU foci at $\gamma$-H2AX-positive loci (Fig. 3f). This observation supports that RAD52 indeed contributes to a DDR-associated MiDAS mechanism, which constitutes a fraction of MiDAS events detected in U2OS cells. Strikingly, RAD51 inhibition significantly reduced both $\gamma$-H2AX-colocalising and non-$\gamma$-H2AX-colocalising EdU foci, indicating that RAD51 promotes MiDAS in a wider context. Furthermore, the inhibition of RAD51, but not RAD52, significantly increased RPA foci (Fig. 3g), indicating that

RAD51 associates with ssDNA, which is otherwise occupied by RPA. Altogether, these data suggest that, while RAD52 plays a specific role in DDR-associated MiDAS by promoting MiDAS downstream of mitotic DNA breaks, RAD51 promotes MiDAS at both DDR-associated and non-DDR-associated loci by binding to ssDNA.

**RAD51 phosphorylation is important for cellular proliferation**. Having established that RAD51 promotes MiDAS, we next sought to elucidate the molecular mechanism by which RAD51 is recruited to mitotic chromatin. In mitotic cells, early DDR events, including $\gamma$-H2AX foci formation, are maintained, but the later events of DDR, such as BRCA1 recruitment, are attenuated[45–47]. Hence, the canonical RAD51 recruitment mechanism mediated

**Fig. 5 Inhibition of RAD51-mediated MiDAS delays anaphase onset in under-replicated cells. a** Schematic diagram of experimental procedure for synchronising HEK293 cells exogenously expressing mCherry-tagged Histone H2B and GFP-tagged Lamin B1. Following double thymidine block and release (dT-B/R) and 5 h RO3306 arrest, cells were released into mitosis (m-R) and mitotic progression was monitored by live microscopy. **b** Mitotic duration of the parental HEK293 cell line, the $RAD51^{S14A/-}$ mutant cell line (S14A) and the wild-type RAD51 complemented cell line (S14A + WT) following cell synchronisation as depicted in **a**. Cells that do not enter into anaphase within 4 h from nuclear envelope breakdown (NEBD) are excluded from analysis. Mitotic duration is measured from NEBD to anaphase onset (AO). Data were obtained from three independent experiments (50 cells analysed per repeat); $n = 150$ per condition. Data distribution is represented by Tukey box-and-whisker plots. Bounds of box are 25–75th percentile, centre shows the median and '+' marks the mean. Whiskers indicate ±1.5xIQR, data outside this range are drawn as individual dots. $p$-values were calculated by a Kruskal–Wallis test followed by uncorrected Dunn's test. Asterisks indicate **$p$-value ≤ 0.01; ****$p$-value ≤ 0.0001. **c** Schematic diagram of experimental procedure for synchronising U2OS cells exogenously expressing mCherry-tagged Histone H2B and GFP-tagged Lamin B1. Following dT-B/R, cells were released into S phase in the absence or presence of 0.4 μM APH ( ±APH-S) followed by 12-h RO3306 arrest. Where indicated, mitotic progression was monitored in the presence of 20 μM RAD51 inhibitor (B02); 2 mM APH (APH-M), 2 μM MPS1 inhibitor (AZ3146), 50 nM neocarzinostatin (NCS) or vehicle (-). All drug treatments were added to cell culture directly after mitotic release (m-R). **d** Mitotic duration of U2OS cells as measured from NEBD to AO. Cells that do not enter into anaphase within 4 h from NEBD are excluded from analysis. Data was obtained from at least three independent experiments; n indicates the total number of cells monitored per condition. Data distribution is represented by Tukey box-and-whisker plots. Bounds of box are 25–75th percentile, centre shows the median and '+' marks the mean. Whiskers indicate ±1.5xIQR, data outside this range are drawn as individual dots. $p$-values were calculated by a Kruskal–Wallis test followed by uncorrected Dunn's test. Asterisks indicate *$p$-value ≤ 0.05; ***$p$-value ≤ 0.001; ****$p$-value ≤ 0.0001. Source data are provided as a Source Data file.

by BRCA2 is likely downregulated in mitosis. Our previous work revealed a non-canonical RAD51 chromatin recruitment mechanism, which is mediated by the cell cycle-dependent phosphorylation of RAD51 at S14 and T13 residues[17,18]. Given that the S14/T13 phosphorylation peaks in mitosis, we hypothesised that the phosphorylation-mediated recruitment of RAD51 supports its mitotic function.

To better understand the impact of physiological RAD51 phosphorylation, we set out to generate U2OS cells in which endogenous *RAD51* gene was edited to encode phospho-mimetic (S14D) or phospho-null (S14A or T13A-S14A) mutant proteins using CRISPR/Cas9 technology (Fig. 4a and Supplementary Fig. 4a, b). As the S14/T13 phosphorylation of RAD51 is strictly sequential (T13 phosphorylation is undetectable for the RAD51 S14A mutant in vivo), the T13A-S14A mutant and S14A mutant are functionally equivalent[18]. Four U2OS clonal lines with homozygous S14D knock-in mutation at the *RAD51* locus, referred to as $RAD51^{S14D/S14D}$, were obtained in the first round of screening as verified by sequencing (Supplementary Fig. 4c, d). However, no positive clones were obtained for the RAD51 phospho-null mutants through two rounds of screening (Supplementary Fig. 4c), suggesting the importance of S14 phosphorylation for U2OS cell viability. Further attempts using other cell lines allowed us to obtain one heterozygous RAD51 S14A mutant in HEK293 cells, referred to as $RAD51^{S14A/-}$, as verified by deep sequencing (Supplementary Fig. 4e). The absence of S14 phosphorylation in the CRISPR gene-edited phospho-mutant cell lines was confirmed by western blot analysis (Supplementary Fig. 4f). Notably, the HEK293 $RAD51^{S14A/-}$ mutant cell line exhibited increased expression of G1-S regulators Cyclin E and p21, but reduced overall activity of PLK1, as detected by phosphorylation at T210 in the PLK1 activation T-loop. These observations indicate a defect in normal cell cycle progression in the $RAD51^{S14A/-}$ mutant cell line. Complementing the HEK293 $RAD51^{S14A/-}$ mutant cell line with the stable exogenous expression of WT RAD51 (S14A + WT) re-established RAD51 S14/T13 phosphorylation. The above-mentioned cell cycle markers in the WT RAD51 complemented $RAD51^{S14A/-}$ cell line remained unaffected, however, indicating that the $RAD51^{S14A/-}$ mutant cells acquired secondary mutations or rewired signalling pathways during the gene-editing process, most likely to alleviate detrimental effects imposed by the loss of RAD51 S14/T13 phosphorylation. Nevertheless, we rationalised that any phenotypes of the $RAD51^{S14A/-}$ mutant cell line that are rescued in the WT RAD51 complemented cell line can be attributed to the loss of RAD51 phosphorylation.

Our earlier studies demonstrated that, in interphase cells, the RAD51 S14A mutant exhibits defects in damage-induced recruitment as well as impaired association with replication forks even in unperturbed conditions[17,18]. Corroborating these observations, HEK293 $RAD51^{S14A/-}$ cells exhibited decreased RAD51 foci formation compared to parental HEK293 cells, which was rescued by WT RAD51 complementation in irradiated cells (Supplementary Fig. 4g). The $RAD51^{S14A/-}$ cell line, compared to the parental HEK293 cell line, also displayed significantly increased sensitivity to low-dose APH (Supplementary Fig. 4h) and reduced cell division in both APH-treated and untreated conditions (Supplementary Fig. 4i). These phenotypes were partially rescued by WT RAD51 complementation (Supplementary Fig. 4h–j). In contrast, we found that the $RAD51^{S14D/S14D}$ mutant cells exhibit enhanced resistance to the radio-mimetic drug neocarzinostatin (NCS) (Supplementary Fig. 4k) and low-dose APH (Supplementary Fig. 4l and m). Taken together, these observations suggest that RAD51 S14/T13 phosphorylation supports cellular proliferation in both HEK293 and U2OS cell lines.

**RAD51 phosphorylation promotes MiDAS.** We next examined our hypothesis that mitotic RAD51 recruitment, and therefore MiDAS, relies on RAD51 S14 phosphorylation. With the CRISPR gene-edited phospho-mutants in hand, we first attempted to assess these cell lines for mitotic EdU incorporation upon mild replicative stress following the synchronisation procedure shown in Fig. 4b. We noticed, however, that the HEK293 $RAD51^{S14A/-}$ mutant cell line displayed an increased G1 population compared to the WT RAD51 complemented $RAD51^{S14A/-}$ cell line (Supplementary Fig. 5a, b), which may ultimately affect the level of under-replication that passes into mitosis. Indeed, while the level of mitotic EdU foci was reduced in the $RAD51^{S14A/-}$ mutant cells compared to the WT RAD51 complemented $RAD51^{S14A/-}$ cell line, this phenotype was accompanied by a significant decrease in mitotic γ-H2AX signal, indicative of reduced mitotic under-replication (Supplementary Fig. 5c–e). Conversely, the $RAD51^{S14D/S14D}$ cell line displayed a markedly faster progression through S phase compared to parental U2OS cells (Supplementary Fig. 5f, g), indicating that the S14D mutation alleviates replication stress introduced during the experimental procedure, likely through mimicking S14 phosphorylation status in S phase when PLK1 activity is intrinsically low. In line with this notion, the $RAD51^{S14D/S14D}$ mutant cells similarly exhibited a significant reduction in mitotic EdU foci (Fig. 4c, f) and both the number and intensity of mitotic γ-H2AX foci (Fig. 4d, e, f). We thus concluded that the severe impact of endogenous RAD51 S14

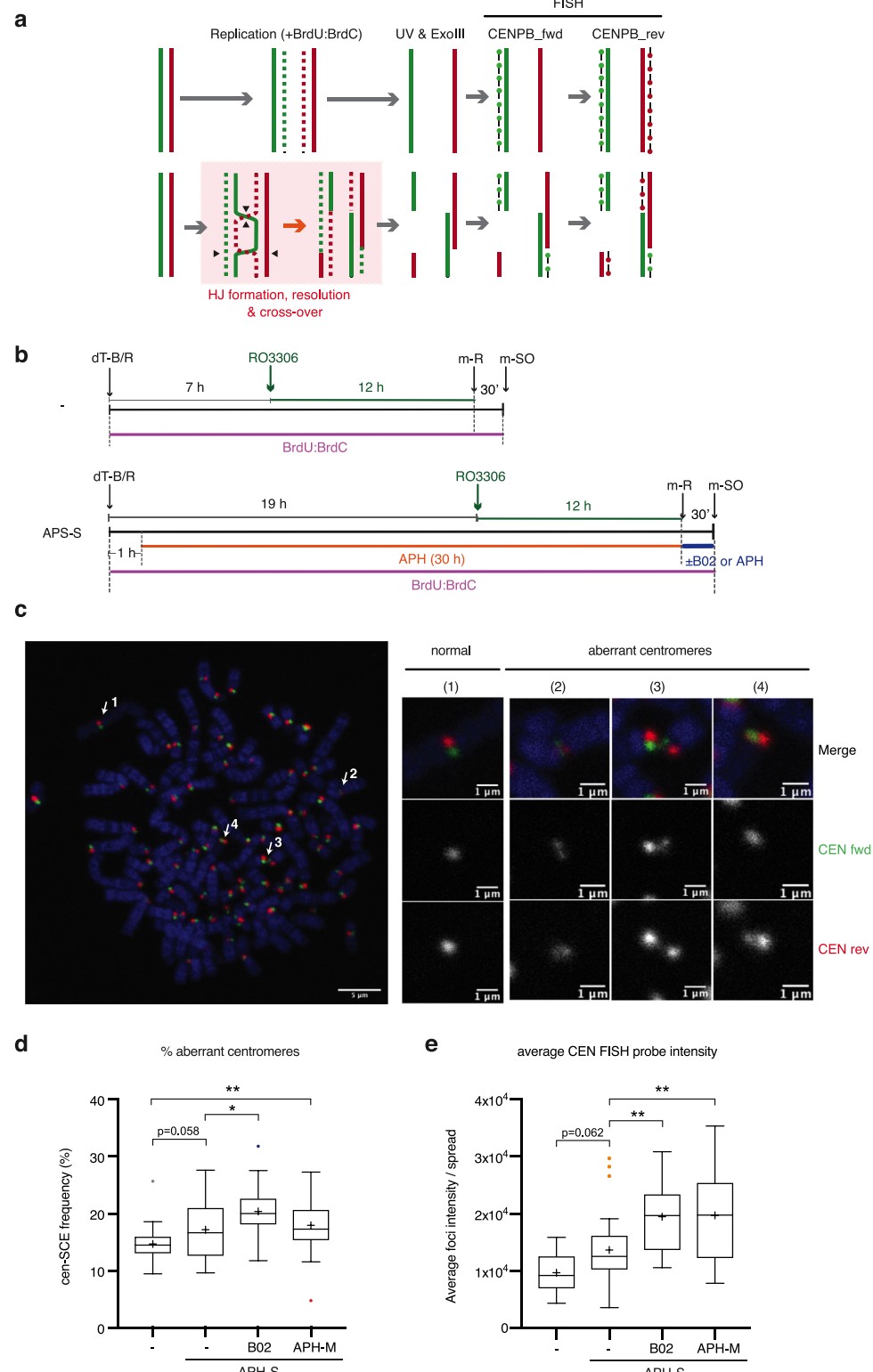

mutation on S phase replication precludes the direct analysis of MiDAS competency in the CRISPR-edited cell lines.

We reasoned that the exogenous expression of RAD51 S14 variants in the presence of endogenous RAD51 may reduce the impact on interphase replication and provide a more straight-forward system by which to assess MiDAS. We therefore assessed mitotic EdU incorporation in previously established U2OS cell lines exogenously expressing RAD51 WT, S14D or S14A, among which

little variation in cell cycle profile was detected[17,18]. Indeed, contrary to CRISPR-edited cell lines, the number of mitotic γ-H2AX foci did not vary between WT RAD51-, S14A- or S14D-expressing cell lines (Fig. 4g and h), confirming similar levels of APH-induced mitotic DNA damage. Markedly, the levels of mitotic EdU foci were significantly reduced in the RAD51 S14A and S14D-expressing cells compared to WT-expressing cells (Fig. 4g, i). Since RAD51 S14D mutation protects cells against replication stress in

**Fig. 6 RAD51 inhibition in mitosis de-protects centromeric DNA. a** Schematic diagram for the experimental procedure of Chromosome Orientation-FISH (CO-FISH). Newly synthesised DNA incorporates BrdU, which is then digested by UV and ExoII treatment. Forward (fwd) and reverse (rev) FISH probes, which hybridise to centromere-specific CENPB-box sequences of the undigested strand, were hybridised in this order. **b** Schematic diagram showing U2OS cell synchronisation, as in Fig. 5c, except cells were exposed to BrdU as cells were released from thymidine block to allow digestion of newly synthesised DNA during the CO-FISH procedure. Cells were then released into mitosis with or without 20 μM RAD51 inhibitor (B02); 2 μM aphidicolin (APH (M)) or vehicle (-). **c** Representative image of CO-FISH. Arrows indicate examples of normal centromere signal (1) and aberrant centromere signal (2–4). Similar images were obtained for three independent experiments. **d, e** Percentage of aberrant centromere signal per metaphase spread obtained in U2OS cells (**d**). The same data was quantified for the average focus intensity of the reverse centromere probe (CEN-rev) per metaphase spread (**e**). Data were obtained from three independent experiments (10 metaphase spreads analysed per repeat); $n = 30$ per condition. Data distribution is represented by Tukey box-and-whisker plots. Bounds of box are 25–75th percentile, centre shows the median and '+' marks the mean. Whiskers indicate ±1.5xIQR, data outside this range are drawn as individual dots. $p$-values were calculated by a Kruskal–Wallis test followed by Dunn's multiple comparison test. Asterisks indicate *$p$-value ≤ 0.05; **$p$-value ≤ 0.01. Source data are provided as a Source Data file.

interphase (Supplementary Figs. 4l, m and 5f, g), we inferred that the reduction of mitotic EdU incorporation in RAD51 S14D-expressing cell lines reflects a decrease in under-replicated loci, which are otherwise targeted by non-DDR-associated MiDAS. In contrast, as RAD51 S14A mutation impairs interphase replication (Supplementary Figs. 4h–j and 5a, b), we regarded the reduction of mitotic EdU incorporation in RAD51 S14A-expressing cell lines as a direct reflection of MiDAS deficiency at under-replicated loci in mitotic cells. Collectively, these results suggest that RAD51 S14/T13 phosphorylation facilitates RAD51 recruitment to under-replicated DNA in mitosis to promote non-DDR-associated MiDAS.

**Under-replicated mitotic cells are sensitive to spindle assembly checkpoint inhibition.** MiDAS takes place upon mitotic entry and before chromosome segregation. As the SAC prevents premature anaphase onset, it simultaneously sets the time window during which the cell may engage in MiDAS. We therefore hypothesised that SAC inhibition limits the time available for cells to complete DNA replication and aggravates the negative consequences of under-replication, resulting in increased cell death. To test this idea, we took advantage of the inherent resistance of U2OS $RAD51^{S14D/S14D}$ cells to replication stress (Supplementary Fig. 4l) and assessed their survival upon exposure to an inhibitor of MPS1 kinase, a central kinase for the maintenance of the SAC. Indeed, compared to the parental U2OS cell line, increased cell survival was observed for the U2OS $RAD51^{S14D/S14D}$ cell line upon MPS1 inhibition, as assessed by cell proliferation (Supplementary Fig. 6a) and, more strikingly, by long-term clonogenic survival (Supplementary Fig. 6b). Conversely, both the HEK293 $RAD51^{S14A/-}$ and WT complemented $RAD51^{S14A/-}$ cell lines, which are highly sensitive to replicative stress (Supplementary Fig. 4h), exhibited significantly increased sensitivity to MPS1 inhibition compared to the parental HEK293 cell line (Supplementary Fig. 6c). Together, these observations suggest that SAC inhibition is more toxic in cells with a high level of replicative stress, hence under-replication.

**Anaphase onset is delayed under conditions of MiDAS inhibition.** The importance of SAC activity for cell survival under replicative stress suggests a potential interplay between the SAC and mitotic under-replication. We hypothesised that the presence of under-replicated DNA in mitosis actively maintains SAC to allow for the completion of DNA replication via MiDAS. To assess the impact of under-replication on SAC maintenance, we monitored mitotic progression in HEK293 $RAD51^{S14A/-}$, which exhibits impaired DNA replication, a phenotype that was rescued upon RAD51 WT complementation (Supplementary Fig. 4i). We measured the time spent from nuclear envelope breakdown (NEBD) to anaphase onset (AO) by live microscopy in synchronised cells expressing GFP-tagged Lamin B1 and mCherry-tagged Histone H2B (Fig. 5a and

Supplementary Fig. 6d). Indeed, HEK293 $RAD51^{S14A/-}$ displayed significantly delayed anaphase onset compared to parental HEK293, which was partially but significantly rescued in the WT RAD51 complemented $RAD51^{S14A/-}$ cell line (Fig. 5b and Supplementary Movies 1–3). This trend was mirrored by the number of mitotic γ-H2AX foci, which was increased in the $RAD51^{S14A/-}$ compared to parental HEK293 cells, and partially rescued in the WT RAD51 complemented $RAD51^{S14A/-}$ cells (Supplementary Fig. 6e, f). Without interphase APH exposure, no significant difference in cell synchronisation was detected between these cell lines (Supplementary Fig. 6g). Given that the loss of S14/T13 phosphorylation increases endogenous replication stress (Supplementary Fig. 4h–j) and impairs MiDAS competency (Fig. 4i), these data indicate a consequential increase in mitotic DNA damage and delayed anaphase onset in under-replicated cells with defective MiDAS.

We further examined whether mitotic duration of U2OS cells could be similarly affected by exogenously induced mild replicative stress, which triggers MiDAS. Indeed, low-dose APH in interphase (APH-S) delayed anaphase onset of U2OS cells (Fig. 5c, d and Supplementary Movies 4 and 5), supporting the notion that replicative stress, arising from either endogenous or exogenous causes, impacts mitotic progression. This phenotype was further enhanced upon MiDAS inhibition by B02 or high-dose APH (APH-M) (Fig. 5d and Supplementary Movies 6–9). U2OS cells exposed to low-dose APH exhibited increased aberrations of mitotic chromatin, such as lagging chromosomes and chromatin bridges (Supplementary Fig. 6h, i) and APH-M treatment in cells exposed to low-dose APH significantly increased mitotic cell death (Supplementary Fig. 6j). The anaphase delay observed in low-dose APH-treated U2OS cells was rescued upon MPS1 inhibition (Fig. 5d and Supplementary Movies 10 and 11), indicating that the SAC mediates metaphase arrest in the presence of under-replication in mitosis. We further found that induction of mitotic DNA breaks upon NCS treatment also triggered a pronounced delay in anaphase onset (Fig. 5d and Supplementary Movie 12). Given that MiDAS inhibition by mitotic B02 or APH-M treatment induces mitotic DDR (Fig. 3d), these observations suggest that the persistence of under-replicated DNA and/or the generation of mitotic DNA damage prevent SAC satisfaction.

**Replication stress affects centromeric integrity in mitosis.** We next investigated the underlying mechanism by which mitotic under-replication delays anaphase onset. The SAC is shown to be activated by DNA damage at centromeres, but not at chromosome arms[48,49], leading us to hypothesise that the centromeric DNA structure might be affected by replicative stress in a manner that prevents satisfaction of the SAC in mitosis. In this scenario, incomplete centromere duplication may delay SAC satisfaction due to the defective kinetochore assembly at each replicated sister

chromatid. Notably, mitotic EdU incorporation and γ-H2AX foci at centromeres was rarely detectable under the conditions tested in this study. However, we reasoned that mitotic EdU incorporation may not be a sufficiently sensitive marker for centromeric under-replication, as the unique cohesin-bound status of centromeric chromatin potentially constrains MiDAS to short tracts that may not be marked by EdU. Similarly, γ-H2AX spreading might be constrained at centromeres. Importantly, centromeres exhibit high levels of sister-chromatid exchange (cen-SCE), detected by chromosome-orientation FISH (CO-FISH) (Fig. 6a), which are elevated under conditions of replicative senescence and intrinsic replication stress[34,35]. We therefore examined whether mild replicative stress and subsequent MiDAS might similarly increase cen-SCE. Our CO-FISH analyses revealed that low-dose APH (APH-S) increased aberrant centromeric signal in U2OS cells, however, this phenotype was not rescued upon MiDAS inhibition by APH-M treatment (Fig. 6b–d). This suggests that mild replication stress induces cen-SCE and/or other structural aberrations, but these events do not entail MiDAS. Counterintuitively, however, we found that B02-mediated mitotic RAD51 inhibition significantly increased aberrant centromeric CO-FISH signals (Fig. 6d). This observation indicates that RAD51 protects centromeric DNA from aberrant recombination events in mitosis. Intriguingly, an increase of the CO-FISH signal intensity was also detected upon APH-S treatment, and this phenotype was further exacerbated by mitotic B02- or APH-M treatments (Fig. 6e). Given that FISH signal intensity reflects the presence of ssDNA to which the FISH probes hybridise, these observations suggest that MiDAS inhibition increased the level of ssDNA at the centromere, either in the context of native centromeric DNA or during the process of sample preparation. Overall, these data suggest that mild replication stress induces centromeric under-replication and leads to the formation of fragile centromeric DNA structures in mitosis. Centromere 'fragility' is exacerbated in the absence of MiDAS and is protected from aberrant recombinogenic processing in a manner dependent on RAD51, hence providing a mechanistic insight into the impact of under-replication on mitotic progression.

## Discussion

This study demonstrates a previously unrecognised role of RAD51 in resolving mitotic under-replication. The role of RAD51 in MiDAS was assessed upon siRNA-mediated RAD51 depletion; addition of a small molecule inhibitor (B02); and in cells expressing the ATPase-defective RAD51 K133R mutant or RAD51 phospho-mutants (S14A and S14D) (Figs. 1–4). In general, the importance of RAD51 in interphase introduces an inherent caveat for the analysis of RAD51 function in MiDAS, as it is difficult to distinguish whether the impact of RAD51 on mitotic EdU incorporation stems from its function during mitosis or instead reflects the indirect consequences of its function in interphase. In this context, the acute inhibition of RAD51 by B02 offered the most reliable method by which the role of RAD51 in mitosis could be studied without affecting the previous interphase. B02 treatment of mitotic cells drastically reduced mitotic EdU incorporation while increasing mitotic γ-H2AX foci; mitotic DNA breaks and mitotic RPA foci (Figs. 2c, d, 3c, d, e and g), indicating a direct role for RAD51 in promoting MiDAS, involving the protection of ssDNA from breakage. Given that B02 specifically inhibits de novo recruitment of RAD51, our findings indicate that RAD51 is actively recruited in mitosis. This is in line with the previous observation that RAD51 is removed by RECQ5 from under-replicated CFSs prior to mitotic entry[50]. Our study also suggests that de novo RAD51 recruitment plays a role at centromeres. We propose that RAD51 is recruited on mitotic chromatin in a PLK1-dependent manner to protect ssDNA, also

at the centromere, which are enriched for PLK1[51]. In this way, RAD51 promotes the completion of DNA replication and regulates subsequent cell division (Fig. 7).

Previous studies have shown that the DDR is largely attenuated in mitosis[46]. However, the early DDR remains intact and the MRN complex is efficiently recruited to mitotic DNA breaks[45,47]. This supports our model that MiDAS is promoted by PLK1-dependent RAD51 S14 phosphorylation, which recruits RAD51 to its target sites by mediating a direct interaction with the MRN complex[18]. Indeed, a previous study reported an active role of PLK1 in promoting MiDAS[24]. PLK1-mediated RAD51 S14 phosphorylation is also promoted by TOPBP1, which has likewise been shown to promote efficient DNA synthesis in mitosis[27,52]. Therefore, it is conceivable that both PLK1 and TOPBP1 impact MiDAS, at least in part, through the recruitment of RAD51. It is noteworthy that the loss of endogenous RAD51 S14 phosphorylation in CRISPR gene-edited RAD51$^{S14A/-}$ mutant greatly affected cell cycle progression and cellular proliferation (Supplementary Fig. 4h–j), indicating that this is a major pathway for RAD51 recruitment even in normal growing conditions without exogenous perturbation. Indeed, it is plausible that the secondary mutations or signalling rewiring in the HEK293 RAD51$^{S14A/-}$ arose as a cellular adaption to the loss of RAD51 phosphorylation. For instance, the overexpression of cyclin E observed in HEK293 RAD51$^{S14A/-}$ cells (Supplementary Fig. 4f) conceivably drives increased origin firing in response to the replication stress induced upon the loss of PLK1-dependent RAD51 recruitment. Interestingly, despite our best effort, we were unable to introduce knock-in mutation for the RAD51 S14A in U2OS cells, potentially indicating the essentiality of PLK1-dependent RAD51 recruitment in this cell line. The importance of RAD51 phosphorylation in different cell lines warrants further investigation.

Earlier studies of MiDAS have focused on its occurrence at FANCD2-marked CFS or telomeres, where MiDAS occurs via a RAD52-dependent break-induced mechanism[30,31]. However, our study reveals the diverse nature of MiDAS depending on the cell line and genomic loci analysed. In particular, our analyses demonstrated that the majority of MiDAS events in U2OS cells occur independently of DNA breaks or DDR, as indicated by the lack of γ-H2AX at sites of mitotic EdU incorporation (Supplementary Fig. 1g and h). Interestingly, while DDR-associated BIR is proposed to occur via conservative replication, a recent study reports that the majority of MiDAS events in U2OS are semi-conservative, supporting the view that MiDAS in U2OS is not entirely a result of BIR[53]. We envisage that these non-DDR-associated MiDAS events occur at stalled forks, which retain the DNA replication machinery and can therefore continue DNA synthesis into mitosis. However, limited EdU incorporation was detectable during RO3306-mediated G2 arrest (Supplementary Fig. 1c), suggesting that the replication of certain difficult-to-replicate loci requires CDK1 activity or indeed, mitotic entry. Several events occur concomitantly with mitotic entry, which may promote the completion of DNA replication. These proposed mechanisms include CDK1-mediated activation of nucleases such as MUS81 to trigger break-induced MiDAS and WAPL-mediated removal of cohesin[24]. Interestingly, WAPL is also proposed to promote RAD51-dependent replication restart at broken forks[4]. However, it is not yet clear how mitotic entry can trigger MiDAS in the absence of fork breakage. The removal of several chromatin-associated proteins in interphase, such as the transcriptional machinery and the cohesin complex at chromosome arms, may prepare the chromatin environment to accommodate MiDAS. Additionally, given that CDKs trigger replication origin firing by phosphorylating the CMG (Cdc45-MCM-GINS) helicase complex, it is also plausible that fully activated CDK1 in mitosis may assist late or dormant origin firing in under-

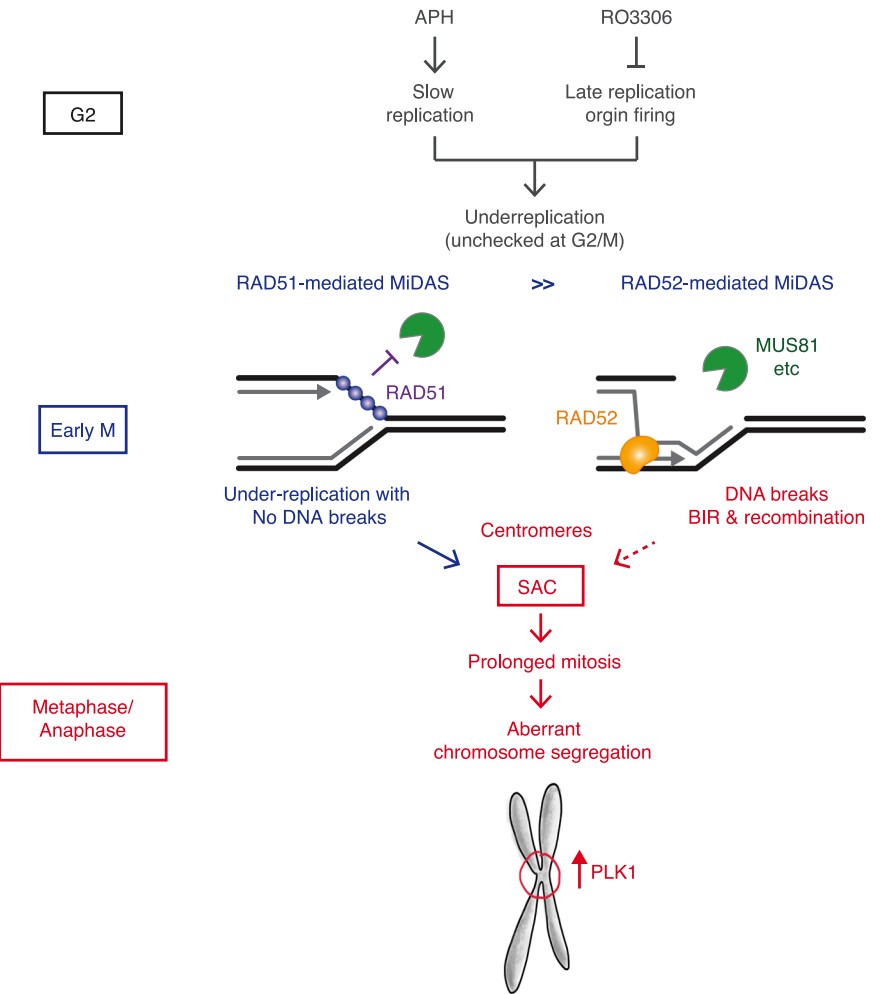

**Fig. 7 Model.** Mild replicative stress (i.e., 0.4 μM aphidicolin) and the suppression of late origin firing (i.e., RO3306) may result in substantial under-replication, which is undetected by the G2/M checkpoint. As cells move into mitosis, RAD51 is recruited to ssDNA at replication forks in a manner dependent on PLK1, and protects them from the attack of mitotic nucleases, such as MUS81, hence allowing the completion of DNA replication. This is distinctive from the RAD52-mediated MiDAS, which occurs downstream of mitotic DNA breaks. Centromeric under-replication prevents SAC inactivation until resolved by MiDAS. RAD51-mediated protection of centromeric DNA, which is enriched with PLK1, hence ensures appropriate chromosome segregation.

replicated regions, which retain the CMG complex. In support of this notion, enriched chromatin association of the CMG complex components were observed in cells exposed to mild replicative stress[24]. Notably, we observed that WEE1 inhibition, which activates CDK1, increased mitotic EdU incorporation (Fig. 1e). While this may simply reflect the mitotic entry of severely under-replicated cells, no significant increase in mitotic γ-H2AX foci was detectable upon WEE1 inhibition (Fig. 1f), conceivably indicating that enhanced CDK1 activity per se promotes break-independent MiDAS.

In agreement with previous reports of a RAD52-dependent BIR-like MiDAS mechanism, we found that RAD52 inhibition prevents the repair of mitotic DNA breaks (Fig. 3e) and specifically promotes DDR-associated MiDAS (Fig. 3f). Indeed, RAD52 depletion increased the accumulation of DNA breaks in mitotic cells without increasing the number of γ-H2AX, supporting the notion that RAD52 acts downstream of DSB induction in mitosis. However, we also found that RAD52-mediated mitotic DNA repair constitutes a relatively small subset of MiDAS events in U2OS cells. In contrast, RAD51 appears to be essential for a broader range of MiDAS events, which take place both at γ-H2AX-positive and negative loci, likely corresponding to damaged and undamaged

forks (Fig. 3f). Indeed, during the preparation of this manuscript, RAD51 was shown to promote break-induced MiDAS at the Fragile X Locus, a folate-sensitive rare fragile site[54]. Considering that nuclease activity is at its highest during mitosis, the protection of DNA intermediates arising from MiDAS is likely vital. Indeed, MUS81 can cleave recombination intermediates formed during BIR, as well as ongoing replication forks, as shown in interphase[55–58]. Notably, we were unable to detect bright RAD51 foci, which are typically associated with damage-induced HR repair, in early mitosis. Given that the role of RAD51 in replication fork protection is not associated with RAD51 foci formation[9], the lack of RAD51 foci observed in mitosis is consistent with a protective function for RAD51 in MiDAS. In line with this notion, enhanced RAD51-mediated DNA protection conferred by ATPase-defective RAD51 K133R expression promotes non-DDR-associated MiDAS (Fig. 2i). Therefore, while RAD51 strand exchange activity may also contribute to MiDAS to some extent, we propose that RAD51 promotes MiDAS primarily through a protective role at the DNA. Of note, our preliminary results indicated that nucleases other than MRE11 are responsible for inducing mitotic DNA damage in the absence of RAD51. This observation hence suggests that RAD51's function in mitosis

is likely distinct from its widely described role in S and G2 in protecting nascent DNA against MRE11-mediated degradation[7,10–12,59]. The detailed mechanism by which RAD51 protects mitotic chromatin merits further investigations in future.

Our study also highlighted the importance of the SAC, which appears to be correlated to the amount of replication stress experienced by the cell. One intriguing possibility is that the SAC may 'sense' mitotic DNA damage or under-replication, thereby preventing SAC silencing until MiDAS has been completed. In line with this idea, we found that APH-induced replication stress during interphase significantly delayed anaphase onset in U2OS cells, and that this delay was dependent on the SAC (Fig. 5d). The APH-induced delay in mitotic progression was further increased upon MiDAS inhibition by either mitotic RAD51 inhibition or mitotic APH treatment. Altogether, these findings suggest that the inability to complete replication in mitosis and/ or the persistence of mitotic DNA breaks prevents SAC inactivation.

The notion that under-replicated DNA or mitotic DNA breaks affect SAC signalling led us to further consider whether these mitotic DNA stresses occur at the centromere. Indeed, mitotic DNA damage at centromeres, but not at chromosome arms, activates SAC[48,49], likely due to the impaired assembly of the functional kinetochore machinery and therefore spindle attachment. Curiously, mitotic EdU incorporation or $\gamma$-H2AX were rarely detectable at centromeres under the conditions we employed, potentially suggesting that centromeric DNA synthesis is inefficient and/or the DDR is downregulated in mitosis at these regions. However, we noticed that conditions of MiDAS inhibition led to an increase in centromeric FISH signal intensity, which we propose is an indication of centromeric 'fragility' (Fig. 6e). Mitotic RAD51 inhibition further triggered an increase in aberrant centromeric CO-FISH signal in cells that experienced mild replication stress (Fig. 6d). Considering the role of RAD51 in protecting under-replicated mitotic chromatin and the fact that centromeres are among the most difficult-to-replicate regions, we envision that centromeres are deprotected upon mitotic RAD51 inhibition and thereby become particularly prone to form non-B DNA structures. It is tempting to speculate that the inherent vulnerability of the centromere to under-replication enables it to act as a gauge for overall genome replication and hence maintains SAC activation to maximise genome replication prior to anaphase. This might be particularly advantageous for organisms that must replicate large genomes and may, in part, explain the complexity of centromeric DNA sequence in eukaryotes.

The identification of players in the MiDAS pathway may reveal promising targets for cancer therapy. This study reveals a role for RAD51 in promoting both break-induced and break-independent MiDAS mechanisms. Furthermore, RAD51 phosphorylation appears to mediate RAD51 recruitment in mitosis. A better understanding into how PLK1-dependent RAD51 phosphorylation is regulated upon DNA damage or replication stress may reveal druggable targets to abrogate RAD51 recruitment in mitosis, and thereby inhibit MiDAS. Lastly, inhibition of SAC activity was shown to severely decrease cell survival in cells that exhibit high levels of replication stress. This suggests that therapies aimed at increasing cellular replication stress may be particularly effective when combined with chemotherapeutic treatments targeting the spindle checkpoint.

## Methods

**Cell culture**. All cell lines were cultured at 37 °C with CO$_2$ in Dulbecco's modified Eagle medium supplemented with 10% v/v fetal bovine serum, streptomycin (0.1 mg/ml) and penicillin (100 units/ml). U2OS Flp-In T-REx cell lines expressing RAD51 variants (Fig. 2 and Supplementary Fig. 2) were generated by transfecting

pcDNA5/FRT encoding FLAG-tagged RAD51 variant to U2OS Flp-In T-REx (a kind gift from Daniel Durocher), followed by hygromycin selection at 200 µg/ml. All transfections were performed using JetPrime (Polyplus transfection). All cell lines used to assess mitotic progression (Fig. 5 and Supplementary Fig. 6) were generated by co-transfecting pcDNA3 encoding mCherry-tagged Histone H2B (Addgene: 20972) and pcDNA3 encoding GFP-tagged Lamin B1 (a kind gift from David Vaux). Stable clones were isolated by FACS. U2OS RAD51 variant expressing cells (Fig. 4) were previously generated[18]. To generate RAD51 S14 knock-in mutant cell lines (Supplementary Figs. 4 and 5), cells were co-transfected with a CRISPR/Cas9 construct (pSpCas9(BB)-2A-GFP) expressing guide RNA targeting the RAD51 S14 locus (AGCAAATGCAGATACTTCAGTGG), as well as the relevant ssDNA repair template (Supplementary Table 1). Clones were screened by PCR (Supplementary Table 2) and further confirmed by DNA sequencing of the RAD51 S14 locus and immunoblotting for RAD51 S14 phosphorylation. The wild-type complemented HEK293 RAD51$^{S14A/-}$ mutant was generated by co-transfecting pcDNA5/FRT encoding wild-type RAD51 and pcDNA-DEST26 (Invitrogen), followed by single cell sorting (FACS) and G418 selection at 400 µg/ml. RAD52 was depleted using ONTargetplus Human RAD52 siRNA (Horizon) and RAD51 was depleted using a mixture of two siRNAs as previously published[18]. MISSION siRNA Universal Negative Control #1 (Sigma-Aldrich) was used as a negative control. All siRNA treatments were performed with 20 nM siRNA using JetPrime (Polyplus transfection).

**Chemical inhibitors**. Where indicated, cells were treated with aphidicolin (Santa Cruz); RO3306 (Cayman Chemical); B02 (Cayman Chemical); AICAR (Sigma-Aldrich); AZD1775 (Selleckchem); AZ3146 (Santa Cruz); Neocarzinostatin (Sigma-Aldrich) at the indicated concentrations and for the indicated lengths of time.

**Cell survival assay**. For clonogenic survival assay, U2OS cells were plated at a density of 40 or 100 cells per well in a 6-well plate. Once adhered, the corresponding drug was added to the cell culture at the indicated dose and cells were incubated for 8 days. Cells were then fixed and stained in 50% methanol, 7% acetic acid and 0.1% Coomassie Brilliant blue. Colonies of >50 cells were counted, and percentage survival was calculated relative to the plating efficiency of the vehicle-treated control. For WST-1 cell proliferation assay, cells were seeded at a density of 5000 cells (U2OS) or 10 000 cells (HEK293) per well in a 96-well plate. Once adhered, the corresponding drug was added to the cell culture at the indicated dose and cells were incubate for 5 days. Cell survival relative to vehicle-treated cells was assessed using the WST-1 kit (Roche) according to the manufacturer's instructions.

**Extract preparation, western blotting, immunoprecipitation**. Cells were collected by trypsinization, washed with ice-cold in PBS and incubated on ice in extraction buffer (50 mM Tris-HCl pH 8.0, 150 mM NaCl, 2 mM EDTA pH 8, 0.5% NP40, 10 mM benzamide hydrochloride, 20 mM NaF, 20 mM $\beta$-glycerophosphate, 1 mM Na$_3$VO$_4$, 5 mM MgCl$_2$, 125 U/ml Benzonase nuclease (Novagen) and 1x Protease inhibitor cocktail (Sigma-Aldrich) for at least 30 min. Supernatant was collected as whole cell extract. For immunoprecipitation, the obtained cell extracts were pre-cleared with uncrosslinked Affiprep Protein-A beads (Bio-Rad) for at least 1 h at 4 °C. The pre-cleared lysate was subsequently incubated with antibody-crosslinked Affiprep Protein-A beads overnight at 4 °C. After extensive washing in extraction buffer, immune complexes were eluted from the beads in NuPAGE LDS sample buffer (ThermoFisher Scientific) supplemented with 12 mM DTT by heating the sample at 85 °C for 5 min. Western blotting was performed following standard protocol. Primary and secondary antibodies were applied at the dilutions described in the Supplemental Methods. Where indicated, the membrane was treated with Re-Blot Plus Mild Solution (Millipore) before incubating with another antibody.

**Flow cytometry**. Cells were collected by trypsinization and fixed in 70% ethanol at approximately 10$^6$ cells/ml. After fixation, cells were permeabilised in PBS with 0.1% Triton X-100 (Sigma-Aldrich) and 1% BSA (Sigma-Aldrich) for 15 min. To detect mitotic cells, the permeabilized cells were incubated for 1 h with anti-phospho-S10 Histone H3 (06-570, Merck Millipore) in PBS containing 0.1% Tween-20 (Sigma-Aldrich) and 1% BSA, followed by a subsequent incubation with Alexa Fluor 488-conjugated secondary antibody (ThermoFisher Scientific) for 30 min. Cells were counter-stained in PBS with 0.1% BSA, 0.1 mg/ml RNAse A (Sigma-Aldrich) and 2 µg/ml propidium iodide (Sigma-Aldrich) for 30 min. Cell cycle distribution was analysed using a FACSCalibur (Becton Dickinson) equipped with CellQuest Pro software (version 6.0); or CytoFLEX LX (Becton Dickinson) equipped with the CytExpert programme (version 2.3.0.84). Obtained datasets were fitted using FlowJo analysis software (versions 10.5.3, 10.6.2 or 10.7.2). Excessive debris were removed from the population through forward versus side scatter (FSC vs SSC) gating where necessary, and single cells were identified through FL2A vs. FL2W gating to discriminate single cells from doublets. Cell counts in G1, S, G2 were estimated using the Watson Pragmatic algorithm in FlowJo on gated interphase cells and combined with quantification of gated mitotic cells to calculate the indicated cell cycle distributions. Gating strategy is depicted in Supplementary Fig. 7.

**Cell synchronisation**. For double thymidine block-and-release, cells were cultured in the presence of 2 mM thymidine (Sigma-Aldrich) for 18 h and released in fresh medium for 9 h, before being incubated in 2 mM thymidine for an additional 17–18 h. Cells were released into fresh medium to allow for progression through S phase. Where indicated, 0.4 μM aphidicolin was added 1 h upon release from the thymidine block. Cells were subsequently arrested in G2 upon the addition of 9 μM RO3306 at the indicated time points and incubated for 5 or 12 h. Cells were released into fresh medium to allow for mitotic entry and collected by mitotic shake-off after 30 min. EdU pulse-labelling was performed at the indicated time points by 30 min incubation in the presence of 10 μM 5-ethynyl-2-deoxyuridine (Life Technologies).

**Immunofluorescence (IF)**. When detecting IF upon irradiation-induced DNA damage, cells were seeded on coverslips one day prior. Where indicated, a Gravitron RX 30/55 (Gravatom) was used to irradiate cell cultures at the stated dose. Cells were returned to cell culture for 3 h post-irradiation and subsequently fixed and permeabilised for 20 min in PTEMF buffer (20 mM PIPES pH 6.8, 10 mM EGTA, 0.2% Triton X-100, 1 mM $MgCl_2$, 4% paraformaldehyde). Primary and secondary antibodies were applied at the dilutions listed in the supplementary methods and incubated at room temperature for at least 1 h. When detecting IF in mitotic shake-off, mitotic cells were seeded onto poly-L-lysine coated coverslips and allowed to adhere for 5–10 min before being fixed and permeabilised for 20 min in PTEMF buffer. Prior to antibody incubation, fixed and permeabilized cells were incubated for 1 h in PBS containing 2 mM CuSO4, 10 mM sodium ascorbate and 10 μM Alexa Fluor azide (ThermoFisher Scientific) to fluorescently label incorporated EdU. Coverslips were washed in PBS with 0.5% Triton X-100 in between incubations and mounted in DAPI-containing ProLong Gold antifade reagent (Life Technologies).

**Pulsed-field gel electrophoresis (PFGE)**. Cells were synchronised and released into the relevant drug for 30 min. Mitotic cells were then collected by mitotic shake-off and 250,000 cells were used per agarose plug, prepared according to manufacturer's instructions (Bio-Rad). Plugs were suspended in proteinase K reaction buffer (100 mM EDTA, pH 8, 1% N-Lauroyl Sarcosine, 0.2% Na-Deoxycholate, 1 mg/ml Proteinase K) overnight, 50 °C, and washed four times in wash buffer (20 mM Tris pH 8, 50 mM EDTA) for 1 h, where 1 mM PMSF was added to the third wash. Electrophoresis was carried out in 0.5X TBE at 14 °C for 15 h using the CHEF Mapper XA system to separate 1–100 kb size DNA bands according to the following settings: two-state, 120° included angle, 6 V/cm, 0.05–10 s switch time, with a linear ramping factor. The gel was stained with ethidium bromide and visualised with a UV transilluminator (UVP). Band intensity was measured in ImageJ and normalised to untreated control.

**Metaphase spreads and CO-FISH**. Cells were synchronised by double thymidine block and released into S phase in the presence of 7.5 μM BrdU and 2.5 μM BrdC. Where indicated, 0.4 μM aphidicolin was added 1 hour upon release from the thymidine block. Cells were subsequently arrested in G2 upon the addition of 9 μM RO3306 at the indicated time points and incubated for 12 h. Cells were released into fresh medium containing 0.1 μg/ml colcemid (Millipore), 7.5 μM BrdU and 2.5 μM BrdC for 30 min. Mitotic cells were subsequently collected by mitotic shake-off and swollen in 0.56% w/v KCl for 20 min at 37 °C. Cells were fixed in methanol:acetic acid (3:1) and dropped onto glass slides. The metaphase spreads were prepared for CO-FISH according to the previously published protocol[35], as outlined in the Supplemental Methods.

**Microscopy analysis and live-cell imaging**. For fixed samples, all images were captured using an Olympus FV1000 Laser Scanning Microscope with Becker and Hickel FLIM system and Olympus Fluoview FV1200 using FV10-ASW software (version 4.2). Foci counting was performed using FIJI (Image J version 2.0.0-rc-65/1.52a or 2.1.0) and the GDSC (FindFoci) ImageJ plugin[60]. Only the foci that overlap with the DAPI staining were analysed. Live-cell images were captured using a Zeiss 880 inverted confocal microscope with Zen Black software (version 14.0.22.201). Images were captured every 4.5 min (Fig. 5b) or 5 min (Fig. 5d) for at least 6 h.

**Statistical analysis**. All statistical analysis was performed using GraphPad PRISM 9 For Mac OS X (Version 9.1.2) with the test indicated in figure legend. Briefly, analyses of multiple groups were conducted primarily by Kurskal–Wallis test. Two-way ANOVA was used for dose curve survival analyses. Analysis of two groups was conducted primarily by Mann–Whitney test. Where appropriate, Brown-Forsythe and Welch ANOVA test, T-test or Welch's T-test was chosen. For p-values that were above 0.05 but <0.1, the exact values are indicated in figures.

**Reporting summary**. Further information on research design is available in the Nature Research Reporting Summary linked to this article.

## Data availability
The data supporting the findings of this study are available from the corresponding authors upon reasonable request. The source data generated and analysed in this study are available in the Open Science Framework (OSF) with the identifier [doi:10.17605/OSF.IO/68X3K][61].

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

## Acknowledgements

We thank Profs Timothy C. Humphrey, Ian D. Hickson, Chris J. Norbury, Dr. James Carrington, and members of the Esashi laboratory for helpful discussions; Drs. Nigel Rust and Michal Maj for assistance with FACS; and Dr. Alan Wainman for assistance with microscopy. F.E. was supported by the Wellcome Trust Senior Research Fellowships in Basic Biomedical Science (101009/Z/13/Z) and is thankful for supports from the Edward Penley Abraham Research Fund. IEW was a recipient of the Medical Sciences Graduate School studentship, funded by the Medical Research Council (14/15_MSD_439771). E.G. is a recipient of the Medical Sciences Graduate School studentship, funded by the Medical Research Council (18/19_MSD_2111222). X.S. receives the Oxford Cancer Centre Cancer Research UK D.Phil studentship (CRUK-OC-DPhil17-XS). L.R. was a recipient of the Oxford Interdisciplinary Bioscience Doctoral Training Partnership, sponsored by the Biotechnology and Biological Sciences Research Council (BB/M011224/1, Project 1757783). A.B. is thankful for the support from the departments of Pathology, Biochemistry, Pharmacology and Physiology, Anatomy and Genetics and from the John Fell Fund and Wellcome ISSF.

## Author contributions

F.E. and I.E.W. conceived and planned the project; I.E.W., E.G., L.R. and F.E. conducted experiments. I.E.W. and L.R. generated CRISPR gene-edited mutants with the help of C.R. and A.B. X.S. contributed to the optimisation of the cen-CO-FISH methodology and the interpretation of the related results. F.E. and I.E.W. wrote the manuscript with input from all contributing authors.

## Competing interests

The authors declare no competing interests.
