## [Peer Review File · Nature Communications]

The RAD51 recombinase protects mitotic chromatin in human cellsReviewers' comments:

Reviewer #1 (Remarks to the Author):

In this manuscript, the authors show that RAD51 is important for efficient MiDAS. B02, an inhibitor of RAD51, reduces EdU foci in mitotic cells significantly. The RAD51 S14A mutant, which cannot be phosphorylated by PLK and function in mitosis, also displays defects in MiDAS. Cells expressing the S14A mutant are defective for mitosis and sensitive to inhibitors of SAC. Moreover, B02 also increases centromere aberrations. The results of this study suggest that RAD51 functions in MiDAS and protects centromeres. However, how RAD51 functions in MiDAS is not clearly explained. The authors suggest that RAD51 protects forks from nucleolytic cleavage because inhibition of RAD51 increases RPA foci in mitosis. However, it is still unclear how RAD51 exerts this function. The manuscript would be much more convincing if the following issues can be addressed.

1. The results in Fig. S1B, C are confusing. EdU foci were detected in G2 cells but disappeared after thymidine chase (G2+Thy). Presumably thymidine chase should not remove EdU from the DNA that is already labeled. Why did the EdU foci in G2 cells disappear? It is also unclear whether the EdU incorporation after m-R occurs in late G2 or early mitosis. Presumably it takes G2 cells some time to enter mitosis. The data in Fig. S1C suggest that most of the EdU incorporation after m-R occurs during the first 5 minutes. Were cells still in late G2 at this time? Did nuclear membrane break down in these cells? Were chromosomes condensed and H3 pSer10 phosphorylated?
2. The data in Fig. S1G and H may be over-interpreted. The formation of gH2AX foci requires spreading of PIKKs on chromatin, which may not occur efficiently at all DSBs on mitotic chromosomes. The lack of gH2AX in mitotic EdU foci does not necessarily mean that replication forks are unbroken.
3. I agree with the authors that the data in Fig. 1 are difficult to interpret because siRAD51 may affect the levels of replication associated damage in S and G2 cells, so its effects on EdU/gH2AX/RPA foci in mitotic cells may be indirect. Given these significant caveats, why did the authors present these data in Fig. 1?
4. The effects of APH(M) in Fig. 2G may be over-interpreted. Although EdU foci were clearly reduced by APH, it is hard to tell whether DNA synthesis is completely blocked. The possibility that gH2AX (DSBs) are generated in mitosis is not ruled out convincingly.
5. The data in Fig. 3 may be over-interpreted. RAD51 wt/T131P/K133R were expressed in cells when they were treated with APH, so the levels of replication associated damage may already be different before cells reached G2. The differences in MiDAS may be indirect effects of the RAD51 mutants rather than direct effects on MiDAS. This is a significant caveat.
6. I was confused by the data in Fig. 4. Why did RAD51 S14D reduce the under-replication in mitotic cells? Presumably under-replication is caused by APH in S phase when RAD51 S14 phosphorylation has not occurred. Even if S14D bypasses PLK and affects cells in S phase, should it also bind replication forks in mitosis more efficiently and increase MiDAS? It is hard to interpret the S14D data if it has opposite effects in S and M phases.
7. I like the data in Fig. 5, S4 and S5 overall, but is it still possible that the protective effects of S14D come from S phase rather than MiDAS? Although the S14A mutant affects mitosis, are these effects caused by defective MiDAS or defective repair in G2? PLK is already active in G2, and the PLK-dependent RAD51 recruitment should work even in cells arrested with RO3306.
8. It would be helpful if the authors can better explain how RAD51 protects replication forks from nucleolytic cleavage during MiDAS. Does RAD51 inhibit nascent DNA degradation and MRN activity in mitosis? If the RAD51-dependent MiDAS occurs at unbroken forks, is this process different from BIR? If so, why is MiDAS completely dependent on RAD52?

Reviewer #2 (Remarks to the Author):

Wassing et al assess the role of RAD51 in mitosis. Specifically, they assert that RAD51 promotes MiDAs, form of mitotic replication, by protecting under-replicated DNA in mitosis. Overall, this is a very descriptive and correlative study of RAD51 function during mitosis and although a lot of complicated experiments are shown, the interpretation of many of the results is difficult.

Major points:

The first major issue is that the authors equate gammaH2AX signal with a presence of a double strand break. It is clear from the literature that this is not the case. If they wanted to substantiate this claim, they would have to show that there was a physical break. Gamma H2AX is increases during replication stress induced by variety of agents when there is no DNA break. At this point they can only say that there is signaling of DNA damage at the gammaH2AX positive sites. Because of that, they cannot draw a conclusion that MiDAS is associated with unbroken DNA in one cell line and broken DNA in a different cell line (page 7) or use the presence or absence of gamma H2AX as an indication of the absence or the presence of breaks in other parts of the manuscript.

The second major issue is the interpretation of experiments with RAD51 mutants. The main problem is lack of complementation of phenotypes in RAD51 S14A cell line (Figure 5). If WT RAD51 does not complement the phenotype, it has to be concluded that the phenotype is due to non-RAD51 related abnormality in that cell line so the cell line cannot should not be used to study RAD51. There is only one clone presented and they indicate that this is the only clone that they obtained. The second issue is that the authors conclude that RAD51S14D mutant has faster progression through S phase. I do not see it in their data. It is also unclear what can be discerned from the use of T131P mutant in the manuscript. In published work, that mutant was shown to form no foci on its own (unlike what the authors state on page 10 and are showing in Figure 3 B). On its own it does not stably associate with the DNA so it is misleading to show it on the ssDNA. It affects the WT RAD51 binding to ssDNA. Also, the expression of this and the KtoR mutant are minuscule in comparison to the WT levels of RAD51 so it is unclear what conclusions can be drawn from these studies.

Other points:

There is a discrepancy between the pS10-H3 levels between Figure 1B and 1C. The siRAD51 condition shows almost no pS10-H3 by western blotting but there are 14.9 % positive cells by flow. The 18.8% positive cells in sicontrol flow correlate with plenty of signal on the western blot.

The writing is often convoluted which makes it difficult to read.

Minor points:

I can see why the authors choose to label the graphs on top but I think that the conventional way of labeling of the Y axes would be better. I find it confusing not to have labels on the Y axes. Also, I keep reading "no." as "no" as in "lack of" and not "#" which again is confusing.

Reviewer #3 (Remarks to the Author):

Midas, or Mitotic DNA synthesis, was described as a protective mechanism upon replication stress. Midas likely results from a break-induced-like replication process that repairs DNA breaks. It was also demonstrated that Midas is RAD52 dependent (with RAD51 being involved only under specific circumstances).

Here, the authors suggest that RAD51 instead promotes Midas, even more than RAD52. It is proposed that RAD51 protects under-replicated DNA from degradation, which is a necessary

condition to facilitate Midas. This function of RAD51 depends on its phosphorylation.

I found the manuscript difficult to read and sometimes confusing. The literature is not always referenced correctly. It is not clear what can explain the differences with respect to the previous studies. Some effects may be due to clonal selection. The explanations sometimes appear ambiguous.

Introduction: Re: "As cells progress into mitosis, however, RAD51 binding to BRCA2 is perturbed by CDK-mediated BRCA2 phosphorylation, suggesting that BRCA2-mediated RAD51 loading to chromatin is downregulated in mitosis (Esashi et al., 2005)." The CDK-mediated phosphorylation only affects RAD51 interaction with the C-terminal BRCA2 site, not with the BRC repeats, which are also important for RAD51 loading. Please clarify.

Fig. 1E-G - It is very difficult to see the differences, change Y axis (?)

Fig. 1D: the image quality is not very good, and does not appear to correspond to the quantitation below. E.g., the gH2AX image appears to show no (or barely detectable) foci in siCntrl, while there appears to be no difference in the quantitation in panel F.

In Wang et al., it is concluded that "RAD51-T131P displays DNA-independent ATPase activity, no DNA pairing capacity and a co-dominant negative effect on RAD51 recombinase function.". In the current manuscript, the mutant is considered deficient in DNA protection but "sustaining HR". That is not correct. The patient cells in Wang et al were proficient in HR, but this was due to the low proportion of T131P to wt RAD51, not because RAD51 T131P supports HR.

Data in Figure 3 are confusing. How can such a small increase of RAD51 (10% over wt level?) result in such a massive (****) stimulation of DNA synthesis? Can the difference be due to clonal variations? The levels of the RAD51 mutants (T131P, K133R), compared to wild type, are barely detectable. Expressing toxic proteins may lead to selection for compensatory mechanisms - are the observed effects due to the expression of the RAD51 mutants or due to potential ill-defined compensatory mechanisms that were selected for? The results should be validated in a transient system.

Figure 4: The data are very difficult to interpret - the authors conclude that phosphorylation at S14D promotes Midas. However, in contrary to that the levels of DNA incorporation in RAD51 SA and SD are identical (decrease of DNA incorporation compared to wild type). While the authors attempt to explain these data by decreased under-replication in SD, I do not think that these results prove that phosphorylation is relevant in this context.

Figure 4A etc. and final Model: It is established that RAD51 - at least in S/G2 cells - protects DNA degradation against MRE11/MRN. The authors previously suggested that MRN recruits phosphorylated RAD51 to DNA. It is difficult to reconcile these conclusions with the model that MRN in mitosis is involved in the recruitment of RAD51 to protect DNA. How can RAD51 protect DNA when MRN is there in the first place?

Response to reviewers' comments

We thank the reviewers for their thorough, scholarly, insightful and constructive comments and we have now incorporated their recommendations and corrections the best we could. Below, we provide a detailed point-to-point response. Please note that the Figure and page numbers referenced by the reviewers in the original manuscript do not match those of the revised manuscript. To facilitate straightforward review, our responses include the relevant sections, Figures, and page numbers for the revised manuscript.

Reviewer #1 (Remarks to the Author):

In this manuscript, the authors show that RAD51 is important for efficient MiDAS. B02, an inhibitor of RAD51, reduces EdU foci in mitotic cells significantly. The RAD51 S14A mutant, which cannot be phosphorylated by PLK and function in mitosis, also displays defects in MiDAS. Cells expressing the S14A mutant are defective for mitosis and sensitive to inhibitors of SAC. Moreover, B02 also increases centromere aberrations. The results of this study suggest that RAD51 functions in MiDAS and protects centromeres. However, how RAD51 functions in MiDAS is not clearly explained. The authors suggest that RAD51 protects forks from nucleolytic cleavage because inhibition of RAD51 increases RPA foci in mitosis. However, it is still unclear how RAD51 exerts this function. The manuscript would be much more convincing if the following issues can be addressed.

1. The results in Fig. S1B, C are confusing. EdU foci were detected in G2 cells but disappeared after thymidine chase (G2+Thy). Presumably thymidine chase should not remove EdU from the DNA that is already labeled. Why did the EdU foci in G2 cells disappear?

We apologise for the confusion that the reviewer might have had on these results. Throughout this paper, we analysed mitotic cells for the presence of EdU foci, which we use as a read-out for mitotic EdU incorporation (i.e. MiDAS). We utilize an experimental set-up in which cells are arrested at the G2/M boundary by RO3306 and, upon mitotic release (m-R), exposed to EdU for 30 minutes. When cells enter mitosis, they round up and can be collected by gentle mitotic shake-off (m-SO). In this way, we separated mitotic cells from cells in interphase and counted the number of EdU foci that overlap with DAPI staining (i.e., chromatin area). These experimental procedures are stated in the main text (page 5) and Method section under 'Cell synchronisation' & 'Microscopy analysis and live cell imaging'. **To clarify that we are exclusively analysing mitotic cells, we have relabelled the y-axis 'EdU foci / mitotic cell'.**

Fig. S1B and C show proof-of-principle results of this approach. Explicitly, we addressed the possibility that EdU incorporation in late G2 could contribute to the presence of EdU foci in mitotically collected cells, hence leading to an overestimation of MiDAS. To test this, we pulsed-labelled cells with EdU for 30 minutes prior to m-R, as depicted in Fig. S1B ('G2'). However, since residual amounts of EdU remain inside of cells even after washing it out from the medium, a subsequent chase with unlabelled thymidine is needed to limit EdU incorporation after m-R ('G2+Thy'). Under these conditions, EdU could only be incorporated prior to m-R, when cells were arrested in G2. Thymidine chase is commonly used for studies of this sort, such as iPOND (Sirbu et al., 2013, Genes Dev.). The observation that EdU foci in mitotically collected cells are drastically reduced in the 'G2+Thy' sample compared to the 'G2' sample demonstrates that the mitotic EdU foci in the 'G2' sample represent incorporation of residual EdU after m-R, and that very little EdU incorporation occurs during the G2-pulse itself. This result therefore supports the notion that the detection of mitotic EdU foci in our experimental set-up can be used as a read-out for MiDAS (i.e. EdU incorporation after m-R). We made this point clear in the revised manuscript (pages 5-6).

It is also unclear whether the EdU incorporation after m-R occurs in late G2 or early mitosis. Presumably it takes G2 cells some time to enter mitosis. The data in Fig. S1C suggest that most of the EdU incorporation after m-R occurs during the first 5 minutes. Were cells still in late G2 at this time? Did nuclear membrane break down in these cells? Were chromosomes condensed and H3 pSer10 phosphorylated?

As discussed in the previous point, we showed that EdU incorporation does not occur prior to release from RO3306 arrest (Fig. S1b and C), at which point cells are poised at the stage defined as 'G2/M checkpoint' (first activated via ATM/ATR activation, followed by CHK1/CHK2 activation, WEE1 activation/CDC25 inhibition, then CDK1 inhibition).

We appreciate that the definition of the G2/M boundary remains a matter of debate in the field of the cell cycle. Canonically, however, M phase is considered to start with mitotic cell rounding and chromosome condensation (early prophase), followed by NEBD (prometaphase), as reviewed by Ramkumar and Baum (Nat. Rev. Mol. Cell Biol., 2016) and The Cell (<https://www.ncbi.nlm.nih.gov/books/NBK9958/>), and depicted in Fig. 5A in this manuscript. As we precisely select for mitotic cell rounding by mitotic shake-off, the collected cells automatically fall within this definition of mitotic entry. We also reason that the events that happen after mitotic entry would affect spindle assembly checkpoint (SAC). Indeed, as shown in Fig. 5D, MiDAS inhibition delays mitotic progression and this is dependent on SAC. In the revised manuscript, we made this point clear in page 13 under 'Under-replicated mitotic cells are sensitive to spindle assembly checkpoint inhibition' and page 14 under 'Anaphase onset is delayed under conditions of MiDAS inhibition'.

2. The data in Fig. S1G and H may be over-interpreted. The formation of γ H2AX foci requires spreading of PIKKs on chromatin, which may not occur efficiently at all DSBs on mitotic chromosomes. The lack of γ H2AX in mitotic EdU foci does not necessarily mean that replication forks are unbroken.

We thank the reviewer for pointing out this issue. We completely agree that γ -H2AX can be formed without DNA double strand breaks – both sensors of DSBs (DNA-PKcs, ATM) and ssDNA (ATR) are able to phosphorylate H2AX at serine 139 (γ -H2AX) at the vicinity of DSB, resected DSB (i.e., ssDNA with 5' end) or stalled replication forks, although extended ssDNA with no breakage has been shown to be defective in activating DDR/checkpoint, presumably enabling mitotic entry of under-replicated cells (Ercilla et al., 2020).

To our knowledge, defects in canonical damage-induced H2AX phosphorylation have not been reported in mitosis – γ -H2AX foci formation is robustly detected in mitotic cells that had been exposed to ionizing radiation or the radiomimetic drug phleomycin (Giunta et al., 2010). Also, mitotic chromatin has been recognised to be highly dynamic (Nozaki et al., 2017). In line with this, we observed a strong increase in γ -H2AX upon radiomimetic drug (NCS) treatment or other drug treatments such as a TOPII inhibitor Etoposide in mitosis (data not included & manuscript in preparation). We thus consider that γ -H2AX-negative MiDAS events occur at unbroken DNA. Regardless, we agree with the reviewer that γ -H2AX-positive EdU foci represent MiDAS events at damaged DNA which may, or may not, be broken. **To clarify this point in the revised manuscript, we categorize MiDAS events as DDR-associated or non-DDR-associated based on γ -H2AX colocalization (pages 6-7).**

In the revised manuscript, we also included Pulsed-field Gel Electrophoresis (PFGE) to assess physical DNA breaks in mitosis (Fig. 3E). We treated cells with RAD51 inhibitor (B02), RAD52 inhibitor (AICAR) and B-family polymerase inhibitor (APH). Mitotically rounded cells were collected by gentle shake off, and subsequently analysed by PFGE. This analysis clearly detected an increase of DNA breaks in cells treated with B02, which also exhibited an increase in γ -H2AX foci (Fig. 3D). These data indicate the enhanced induction of DNA breakage in mitosis in the absence of RAD51.

In cells treated with AICAR, we also observed an increase of DNA breaks by PFGE (Fig. 3E), but mitotic γ -H2AX foci remained unchanged (Fig. 3D). We believe that these observations are in line with the previously proposed model that RAD52 promotes MiDAS via a BIR-like mechanism at stalled replication forks which are actively cleaved upon mitotic entry (Bhowmick et al., 2016). It is important to note that γ -H2AX foci persist for several hours after DNA repair (Barnard et al., 2013). Since we collect mitotic cells 30 minutes after m-R, γ -H2AX foci at mitotic DNA breaks are expected to persist regardless of break repair. Therefore, a defect in BIR-like MiDAS would lead to the accumulation of unrepaired mitotic DNA breaks, as detected by PFGE, without impacting mitotic γ -H2AX foci. The phenotypes observed upon AICAR treatment therefore suggest that RAD52 promotes

BIR-like MiDAS at broken forks. Indeed, we also find that RAD52 inhibition exclusively reduces γ -H2AX-positive EdU foci (Fig. 3F), further corroborating that break-induced MiDAS events colocalise with γ -H2AX.

Taken together, we conclude that RAD51 plays a pivotal role in protecting DNA from breakage in mitosis, while RAD52 promotes repair downstream of broken DNA. **These points are articulated in the revised manuscript pages 9-10 and our models are depicted in Fig S3A.**

3. I agree with the authors that the data in Fig. 1 are difficult to interpret because siRAD51 may affect the levels of replication associated damage in S and G2 cells, so its effects on EdU/gH2AX/RPA foci in mitotic cells may be indirect. Given these significant caveats, why did the authors present these data in Fig. 1?

We thank the reviewer for acknowledging the challenge in assessing RAD51 function using siRNA. Nonetheless, this approach is commonly used in the field. To compare the conclusions in this paper with what has previously been demonstrated in other published work, we feel it is important to show the results using siRNA targeting RAD51.

4. The effects of APH(M) in Fig. 2G may be over-interpreted. Although EdU foci were clearly reduced by APH, it is hard to tell whether DNA synthesis is completely blocked. The possibility that gH2AX (DSBs) are generated in mitosis is not ruled out convincingly.

It is well established that 2 mM APH, the condition we used for APH(M), is sufficient to stop DNA synthesis. Indeed, this concentration of APH is commonly used for cell cycle synchronisation at the G1/S boundary. To our knowledge, there is no report indicating that APH is less effective on mitotic chromatin. Conversely, 2 mM APH has been used to inhibit MiDAS by Ian Hickson and others in previous work (e.g. Minocherhomji et al., Nature, 2016). Indeed, APH(M) treatment does not decrease γ -H2AX (Fig 3D), suggesting that MiDAS is not a major source of mitotic DNA damage or DDR signalling. We therefore propose that, in the presence of RAD51-mediated DNA protection, forks can continue to progress in mitosis without breakage.

5. The data in Fig. 3 may be over-interpreted. RAD51 wt/T131P/K133R were expressed in cells when they were treated with APH, so the levels of replication associated damage may already be different before cells reached G2. The differences in MiDAS may be indirect effects of the RAD51 mutants rather than direct effects on MiDAS. This is a significant caveat.

We completely agree with the reviewer – the impact of RAD51 mutations in interphase complicates the interpretation of their effect in MiDAS, hence we tried to be very conscientious of this in the original manuscript. In the revised manuscript, we have made the following amendments:

(1) We removed the T131P mutant from our analysis, as we doubt that the level of exogenous expression achieved in this cell line is sufficient to confer the dominant negative phenotype previously reported (Wang et al. 2015).

(2) We analysed the impact of the doxycyclin-inducible expression of RAD51 wt or K133R on MiDAS. This allowed us to induce RAD51 expression specifically during RO3306 arrest (i.e. G2) in order to minimize the impact during the earlier phase of the cell cycle. We also made sure to optimise doxycycline exposure to achieve equivalent levels of RAD51 expression between cell lines (Fig. 2E and F).

The newly presented result demonstrates that the exogenous expression of RAD51 K133R significantly increased non-DDR-associated MiDAS (Fig. 2I) and reduces the formation of mitotic γ -H2AX foci compared to wt RAD51 expression (Fig S2F). Given that K133R is best characterised as an ATPase-deficient RAD51 variant that promotes its stable ssDNA association, this suggests that enhanced RAD51 binding at under-replicated regions protects them from breakage to promote non-DDR-associated MiDAS. We believe that this data complements our results

using B02, which conversely blocks RAD51 binding to ssDNA, and strengthens our conclusion that RAD51 association with ssDNA promotes MiDAS. These points are described in page 8 in the revised manuscript.

6. I was confused by the data in Fig. 4. Why did RAD51 S14D reduce the under-replication in mitotic cells? Presumably under-replication is caused by APH in S phase when RAD51 S14 phosphorylation has not occurred. Even if S14D bypasses PLK and affects cells in S phase, should it also bind replication forks in mitosis more efficiently and increase MiDAS? It is hard to interpret the S14D data if it has opposite effects in S and M phases.

We apologise for any confusion caused. While RAD51 phosphorylation is normally cell cycle-regulated and peaks in mitosis, the S14D mutation mimics constitutive PLK1 phosphorylation throughout the cell cycle. It stands to reason that the impact of the S14D mutation will be minimal during mitosis, when wt RAD51 phosphorylation is naturally at its highest, and greater during S phase, when RAD51 phosphorylation is naturally very low. Indeed, we propose that the enhanced association of RAD51 S14D during S phase reduces replicative stress, thereby reducing the need for MiDAS in early mitosis. This is line with our observation that S14D is more resistant to MPS1 inhibition. We clarified these points in the revised manuscript (pages 12-13 under the section 'RAD51 phosphorylation promotes MiDAS', and pages 13-14 under 'Under-replicated mitotic cells are sensitive to spindle assembly checkpoint inhibition').

7. I like the data in Fig. 5, S4 and S5 overall, but is it still possible that the protective effects of S14D come from S phase rather than MiDAS? Although the S14A mutant affects mitosis, are these effects caused by defective MiDAS or defective repair in G2? PLK is already active in G2, and the PLK-dependent RAD51 recruitment should work even in cells arrested with RO3306.

As above, we agree with the reviewer that the protective effect of S14D comes from S phase. Accumulating evidence over recent years has demonstrated that cells can move into mitosis in the presence of ssDNA and/or incomplete replication, but to our knowledge, there is no convincing evidence to indicate that this is also the case for DNA breaks (i.e. bypass of G2/M checkpoint when DNA breakages remain unrepaired). In the revised manuscript, we made this point clear (page 14 under 'Under-replicated mitotic cells are sensitive to spindle assembly checkpoint inhibition').

8. It would be helpful if the authors can better explain how RAD51 protects replication forks from nucleolytic cleavage during MiDAS. Does RAD51 inhibit nascent DNA degradation and MRN activity in mitosis? If the RAD51-dependent MiDAS occurs at unbroken forks, is this process different from BIR? If so, why is MiDAS completely dependent on RAD52?

Our results presented in this manuscript refute that MiDAS is completely dependent on RAD52. Rather, they support the notion that RAD52 contributes to a subset of MiDAS events, which take place downstream of mitotic DNA breaks. We found that RAD52 inhibition specifically reduces MiDAS events that colocalise with γ -H2AX, which we have termed DDR-associated MiDAS. Additionally, as described previously (see point 2), cells treated with RAD52 inhibitor exhibited an increase of mitotic DNA breaks compared to DMSO control, indicating the persistence of unrepaired mitotic DNA breaks in the absence of RAD52 activity (Fig. 3E). In contrast, RAD51 inhibition reduces both DDR-associated and non-DDR-associated MiDAS, indicating that MiDAS, in general, is more dependent on RAD51 than RAD52. In the revised manuscript, we made the differential contribution of RAD51 and RAD52 in MiDAS clear (Fig. 3F).

In line with previously proposed models of MiDAS, we propose that a subset of under-replicated regions undergo replication fork breakage upon mitotic entry and subsequently engage in BIR-like MiDAS, which is RAD52 dependent. Conversely, we propose that RAD51 associates with ssDNA, either at replication forks or at BIR-intermediates, and in this way, promotes both DDR-associated (i.e. BIR-mediated) and non-DDR-associated MiDAS. **Our model is shown in Fig. 7 in the revised manuscript.** We appreciate the Reviewer's comment as to whether MRN contributes to nascent DNA degradation in mitosis. Our preliminary results indicate that MRE11 inhibition does not rescue RAD51 inhibition phenotypes, and we believe that other nucleases are involved in this

process (discussed in the revised manuscript, page 20). The detail mechanisms by which RAD51 protect mitotic chromatin remain areas of active investigation in my laboratory and are far beyond the scope of this manuscript.

Reviewer #2 (Remarks to the Author):

Wassing et al assess the role of RAD51 in mitosis. Specifically, they assert that RAD51 promotes MiDAs, form of mitotic replication, by protecting under-replicated DNA in mitosis. Overall, this is a very descriptive and correlative study of RAD51 function during mitosis and although a lot of complicated experiments are shown, the interpretation of many of the results is difficult.

Major points:

The first major issue is that the authors equate gammaH2AX signal with a presence of a double strand break. It is clear from the literature that this is not the case. If they wanted to substantiate this claim, they would have to show that there was a physical break. Gamma H2AX is increases during replication stress induced by variety of agents when there is no DNA break. At this point they can only say that there is signaling of DNA damage at the gammaH2AX positive sites. Because of that, they cannot draw a conclusion that MiDAS is associated with unbroken DNA in one cell line and broken DNA in a different cell line (page 7) or use the presence or absence of gamma H2AX as an indication of the absence or the presence of breaks in other parts of the manuscript.

We thank Reviewer 2 for their suggestion to assess physical DNA breaks. We now included a PFGE result to detect physical DNA breaks in mitotic cells (Fig. 3E) and also amend our definitions of γ -H2AX positive and negative MiDAS as DDR- or non-DDR associated (pages 6-7). Please find our response to the **reviewer 1, point 2**.

The second major issue is the interpretation of experiments with RAD51 mutants. The main problem is lack of complementation of phenotypes in RAD51 S14A cell line (Figure 5). If WT RAD51 does not complement the phenotype, it has to be concluded that the phenotype is due to non-RAD51 related abnormality in that cell line so the cell line cannot should not be used to study RAD51. There is only one clone presented and they indicate that this is the only clone that they obtained.

We appreciate the reviewer's concerns and agree that RAD51 S14A survivals upon drug exposure are not always rescued by WT RAD51 complementation (Fig. S6C). Nonetheless, we observed a number of phenotypes which are (partially) rescued by WT RAD51 complementation, and we believe it is justifiable to attribute these to RAD51 S14A mutation. These include: impaired formation of IR-induced RAD51 foci (Fig. S4G), increased sensitivity to APH (Fig. S4H-J), reduced cell division index (Fig. S4I), delayed S-phase entry (Fig. S5B) and delayed mitotic progression (Fig. 5B).

Unfortunately, we had trouble obtaining additional S14A clones, likely due to the negative impact of this mutation for cell survival. To address this challenge, we made sure to use several different approaches to study the impact of RAD51 in mitosis.

The second issue is that the authors conclude that RAD51 S14D mutant has faster progression through S phase. I do not see it in their data.

We apologise for the confusion and agree with the reviewer that this is not obvious in the U2OS cell line which exogenously expresses RAD51 S14D. We found that this phenotype is much clearer in the CRISPR-edited endogenous S14D mutant (i.e. quicker progression through interphase leads to earlier accumulation of S14D cells in the subsequent G1 phase). We have included this analysis of the CRISPR-edited S14D mutant in the revised manuscript (Fig. S5G). We noticed that S14D cells enter S-phase immediately upon dT-B/R, hence forming a larger S phase population (78.6%) compared to parental cells (31.4%). From FACS time point 1 (dT-B/R) to point 2 (interphase upon 0.4 μ M APH exposure), G2 population of S14D mutant also exhibited a marked

increase from 0.5% to 44%, while parental cells increased from 4% to 29.5%. In line with these results, we also found that the CRISPR-edited S14D mutant exhibits increased resistance to low-dose APH (Fig. S4L and M).

It is also unclear what can be discerned from the use of T131P mutant in the manuscript. In published work, that mutant was shown to form no foci on its own (unlike what the authors state on page 10 and are showing in Figure 3 B). On its own it does not stably associate with the DNA so it is misleading to show it on the ssDNA. It affects the WT RAD51 binding to ssDNA.

We apologise our erroneous description. Of note, we decided to remove the T131P mutant from our analysis as addressed to **reviewer 1, point 5**.

Also, the expression of this and the K to R mutant are minuscule in comparison to the WT levels of RAD51 so it is unclear what conclusions can be drawn from these studies.

We thank the reviewer for this comment. In the revised manuscript, we include the analysis of MiDAS upon induced expression of wt RAD51 or RAD51 K133R (Figs. 2G-I, S2D-F). Careful optimisation of these conditions allowed us to achieve comparable levels of expression for both RAD51 variants (see response to the **reviewer 1, point 5**).

Other points:

There is a discrepancy between the pS10-H3 levels between Figure 1B and 1C. The siRAD51 condition shows almost no pS10-H3 by western blotting but there are 14.9 % positive cells by flow. The 18.8% positive cells in sicontrol flow correlate with plenty of signal on the western blot.

We apology for this error – we realised that cells we used for the western blot in Fig 1B were not synchronised. We have now replaced Fig. 1B with a western blot of cells that were synchronisation under the same conditions as cells analysed in Fig. 1C. The levels of pS10 by western blotting now match those detected by flow cytometry.

The writing is often convoluted which makes it difficult to read.

We are very sorry to hear this. In the revised manuscript, we have made it a priority to simplify our writing style.

Minor points:

I can see why the authors choose to label the graphs on top but I think that the conventional way of labeling of the Y axes would be better. I find it confusing not to have labels on the Y axes. Also, I keep reading “no.” as “no” as in “lack of” and not “#” which again is confusing.

We thank the reviewer for this comment and have followed their suggestions to clarify graph labelling.

Reviewer #3 (Remarks to the Author):

Midas, or Mitotic DNA synthesis, was described as a protective mechanism upon replication stress. Midas likely results from a break-induced-like replication process that repairs DNA breaks. It was also demonstrated that Midas is RAD52 dependent (with RAD51 being involved only under specific circumstances).

Here, the authors suggest that RAD51 instead promotes Midas, even more than RAD52. It is proposed that RAD51 protects under-replicated DNA from degradation, which is a necessary condition to facilitate Midas. This function of RAD51 depends on its phosphorylation.

I found the manuscript difficult to read and sometimes confusing. The literature is not always referenced correctly. It is not clear what can explain the differences with respect to the previous studies. Some effects may be due to clonal selection. The explanations sometimes appear ambiguous.

We apologise the complexity of our original manuscript. In the revised manuscript, we significantly changed our narrative in a way that we believe is more straightforward.

Introduction: Re: " As cells progress into mitosis, however, RAD51 binding to BRCA2 is perturbed by CDK-mediated BRCA2 phosphorylation, suggesting that BRCA2-mediated RAD51 loading to chromatin is downregulated in mitosis (Esashi et al., 2005)." The CDK-mediated phosphorylation only affects RAD51 interaction with the C-terminal BRCA2 site, not with the BRC repeats, which are also important for RAD51 loading. Please clarify.

We appreciate the confusion. The reviewer is correct that BRCA2 phosphorylation does not totally block its interaction with RAD51, but impairs/alters it. To simplify our message, we have removed this description in the revised manuscript. Regardless, the downregulation of canonical RAD51 recruitment via the BRCA1-PALB2-BRCA2 axis has been demonstrated in mitosis, and this is highlighted in the relevant result section, pages 10-11 under 'RAD51 phosphorylation is important for cellular proliferation'.

Fig. 1E-G - It is very difficult to see the differences, change Y axis (?)

We thank the reviewer for pointing this out. We have changed the way we plot our data in a way which we hope, makes it easier to see the difference.

Fig. 1D: the image quality is not very good, and does not appear to correspond to the quantitation below. E.g., the gH2AX image appears to show no (or barely detectable) foci in siCntl, while there appears to be no difference in the quantitation in panel F.

We have replaced representative images where necessary to make sure they reflect the corresponding quantifications.

In Wang et al., it is concluded that "RAD51-T131P displays DNA-independent ATPase activity, no DNA pairing capacity and a co-dominant negative effect on RAD51 recombinase function.". In the current manuscript, the mutant is considered deficient in DNA protection but "sustaining HR". That is not correct. The patient cells in Wang et al were proficient in HR, but this was due to the low proportion of T131P to wt RAD51, not because RAD51 T131P supports HR.

We thank the reviewer for pointing out our error. Our revised manuscript no longer includes analysis of the T131P mutant (see **Reviewer 1, point 5**).

*Data in Figure 3 are confusing. How can such a small increase of RAD51 (10% over wt level?) result in such a massive (****) stimulation of DNA synthesis? Can the difference be due to clonal variations? The levels of the RAD51 mutants (T131P, K133R), compared to wild type, are barely detectable. Expressing toxic proteins may lead to selection for compensatory mechanisms - are the observed effects due to the expression of the RAD51 mutants or due to potential ill-defined compensatory mechanisms that were selected for? The results should be validated in a transient system.*

See our response to **Reviewer 1, point 5**. In our revised manuscript, we follow Reviewer 3's suggestion to analyse MiDAS upon conditional induction of RAD51 variants. We made induced expression of wt and K133R RAD51 variants during RO3306 arrest to achieve comparable levels of RAD51 and re-assessed EdU foci associated with DDR (γ -H2AX positive loci) or non-DDR (γ -H2AX negative loci). The resultant data suggests that

RAD51 association with ssDNA supports MiDAS at non-DDR associated loci. These are described in page 8 and Figs. 2E-I, S2D-F in the revised manuscript.

Figure 4: The data are very difficult to interpret - the authors conclude that phosphorylation at S14D promotes Midas. However, in contrary to that the levels of DNA incorporation in RAD51 SA and SD are identical (decrease of DNA incorporation compared to wild type). While the authors attempt to explain these data by decreased under-replication in SD, I do not think that these results prove that phosphorylation is relevant in this context.

We appreciate the complication of results presented. Indeed, mitotic EdU incorporation is determined not only by the proficiency of MiDAS, but also by the level of under-replication that remains in mitosis. Given the impact of S14D in interphase (i.e. reduction under-replication), we agree with the reviewer that we cannot draw conclusions about the impact of S14D on MiDAS proficiency, and we do not claim this. Instead, we believe that the analysis of the RAD51 S14A-expressing cells reveals the importance of RAD51 S14 phosphorylation for MiDAS. Given that the RAD51 S14A mutation confers increase vulnerability to replication stress (hence increased under-replication), the fact that mitotic EdU incorporation is decreased indicates impaired MiDAS proficiency.

Please also see our response to **Reviewer 1, point 6**. We have revised our text to make this clear (pages 12-13 under the section 'RAD51 phosphorylation promotes MiDAS').

Figure 4A etc. and final Model: It is established that RAD51 - at least in S/G2 cells - protects DNA degradation against MRE11/MRN. The authors previously suggested that MRN recruits phosphorylated RAD51 to DNA. It is difficult to reconcile these conclusions with the model that MRN in mitosis is involved in the recruitment of RAD51 to protect DNA. How can RAD51 protect DNA when MRN is there in the first place?

The MRN complex has been shown to travel together with replication machinery, as detected by iPOND (Sirbu et al., 2011), such that the presence of the MRN-complex does not necessarily imply fork degradation is happening. In the meantime, our preliminary results show that MRE11 inhibition does not rescue phenotypes associated with RAD51 inhibition, suggesting that RAD51's action in mitosis is not necessarily linked with MRE11. We have discussed this in the revised manuscript (page 20, the last paragraph under Discussion 'Mechanism mediating MiDAS') and revised the model accordingly (Figure 7).

Reviewers' comments:

Reviewer #1 (Remarks to the Author):

The authors have significantly improved the paper. I think that manuscript is now acceptable for publication.

Reviewer #2 (Remarks to the Author):

This is a much improved manuscript from the standpoint of readability and clarity of data presentation. Some issues remain including analysis of single clones for assessment of the phospho RAD51 mutations. This continues to be a descriptive study and the issue of the lack of clear mechanism remains. However, as pointed out by the authors, future studies should address the mechanism.

For PFGE gel, was the total DNA measured in each lane to quantify the % broken DNA? There could be substantial differences in the DNA in each plug leading to changes in quantification of the physical breaks.

Labeling on graphs is greatly improved. However, "pntI" is not a common abbreviation. please use "parental" or just "WT" U2OS as a label.

Reviewer #3 (Remarks to the Author):

The authors have generally clarified the manuscript and replied to my comments. The finding that RAD51 protects mitotic DNA to allow Midas is interesting and relevant. I believe the paper will be of interest to the community and I will look forward to seeing it published.

We are delighted to hear that all reviewer found our manuscript to be significantly improved and of great interest for the audience of Nature Communication. Please find our point-by-point response below:

REVIEWERS' COMMENTS

Reviewer #1 (Remarks to the Author):

The authors have significantly improved the paper. I think that manuscript is now acceptable for publication.

We are delighted to hear the comment. The previous comments significantly helped us to improve the manuscript.

Reviewer #2 (Remarks to the Author):

This is a much improved manuscript from the standpoint of readability and clarity of data presentation. Some issues remain including analysis of single clones for assessment of the phospho RAD51 mutations. This continues to be a descriptive study and the issue of the lack of clear mechanism remains. However, as pointed out by the authors, future studies should address the mechanism.

We appreciate the reviewer's comment. Indeed, there are numbers of new question raised from this study, and we are looking forward to finding out more about the role of RAD51 in mitosis.

For PFGE gel, was the total DNA measured in each lane to quantify the % broken DNA? There could be substantial differences in the DNA in each plug leading to changes in quantification of the physical breaks.

As described in the method section for PFGE, the same number of cells from each condition are embedded in each plug. This is a standard procedure to preserve the DNA status in cells. Hence, we infer that a similar level of DNA is present in each plug. Additionally, for the analysis and fold-change displayed on the figure, we calculated a ratio between intact DNA (remaining in well) and broken DNA (fast migrating form) before normalising to the DMSO control, further minimizing the impact of any potential differences in DNA levels across the samples.

Labeling on graphs is greatly improved. However, "pntl" is not a common abbreviation. please use "parental" or just "WT" U2OS as a label.

As suggested, we have amended it to 'parental' throughout the manuscript and in figures to be explicit.

Reviewer #3 (Remarks to the Author):

The authors have generally clarified the manuscript and replied to my comments. The finding that RAD51 protects mitotic DNA to allow Midas is interesting and relevant. I believe the paper will be of interest to the community and I will look forward to seeing it published.

We are pleased to hear the comment. We thank again for the comments which had helped us improve the manuscript.